# A kinematic synergy for terrestrial locomotion shared by mammals and birds

Giovanna Catavitello[1,2]*, Yury Ivanenko[2], Francesco Lacquaniti[1,2,3]

[1]Centre of Space Bio-medicine, University of Rome Tor Vergata, Rome, Italy;
[2]Laboratory of Neuromotor Physiology, IRCCS Santa Lucia Foundation, Rome, Italy;
[3]Department of Systems Medicine, University of Rome Tor Vergata, Rome, Italy

**Abstract** Locomotion of tetrapods on land adapted to different environments and needs resulting in a variety of different gait styles. However, comparative analyses reveal common principles of limb movement control. Here, we report that a kinematic synergy involving the planar covariation of limb segment motion holds in 54 different animal species (10 birds and 44 mammals), despite large differences in body size, mass (ranging from 30 g to 4 tonnes), limb configuration, and amplitude of movements. This kinematic synergy lies at the interface between the neural command signals output by locomotor pattern generators, the mechanics of the body center of mass and the external environment, and it may represent one neuromechanical principle conserved in evolution to save mechanical energy.
DOI: https://doi.org/10.7554/eLife.38190.001

## Introduction

Terrestrial locomotion of animals has evolved in vastly different designs adapted to the specific habitat of each species (*Hildebrand, 1976*; *Grillner, 1981*). Anatomically, tetrapods may differ in the number of limbs used for locomotion (bipedal versus quadrupedal), limb length, shape, and mass. Functionally, locomotor styles may differ in terms of limb posture (more flexed or more extended), duty factor (percent of stride interval when each hind foot is on the ground), and for quadrupeds diagonality (percent of stride interval that a forefoot lags behind ipsilateral hind foot). Nevertheless, there might be general principles of organization that underlie the diversity of locomotor styles (*Blickhan and Full, 1993*).

Following the pioneering program set up by Marey, one important research goal in the study of comparative physiology of movement is 'to point out the laws which are common for all forms and manifestations of locomotion' (*Marey, 1874*). Although this ambitious goal has not been reached yet, some general principles for terrestrial locomotion have emerged that apply to a wide range of animal species, mainly related to energy saving mechanisms (*Alexander, 1989*; *Dickinson et al., 2000*) on the one hand, and to the neural control of muscle activity patterns (*Lacquaniti et al., 2013*; *Grillner, 2018*) on the other hand.

With regard to energy saving mechanisms, the exchange between the gravitational potential energy and the forward kinetic energy due to pendulum-like oscillations of the centre of body mass (COM) has been shown to apply to walking of several legged animals (*Cavagna et al., 1977*). With regard to neuromuscular control, four main activity patterns output by spinal motoneurones have been described that are common to different mammals and birds (*Dominici et al., 2011*; *Wenger et al., 2016*). They contribute to the different phases of gait cycle, that is limb extension at foot touch-down, body-weight support during stance, limb lift-off, and swing (*Ting et al., 2015*). Also the architecture of the central pattern generators is highly conserved across animal species (*Kiehn, 2016*; *Grillner, 2018*).

*For correspondence:
g.catavitello@hsantalucia.it

Competing interests: The authors declare that no competing interests exist.

**eLife digest** Animals have evolved very different body shapes and styles of movement that are adapted to their needs in the habitats they live in. For example, mice, lions and many other animals use four limbs to walk, while humans and birds only use two limbs.

The styles animals use to walk also differ in terms of how long each foot is on the ground during a single stride, and for four-legged animals, in how long a forefoot lags behind the hindfoot on the same side of the body during the stride. Yet, there are general principles in how walking is organized that are shared between animals of vastly different shapes and sizes. Many animals save energy during walking by swinging the center of their body mass back and forth like a pendulum.

Networks of neurons are responsible for controlling how and when animals move, and these networks have similar architectures and patterns of activity in many different mammals and birds. How do signals from the nervous system regulate the position of the center of body mass while an animal walks?

Here, Catavitello et al. addressed this question by analyzing how over 50 different species of birds and mammals walked around in zoo enclosures and other semi-natural or natural environments. The species studied ranged in size from mice weighing around 30 grams to elephants weighing around 4 tonnes. The team also studied human volunteers walking on treadmills.

The experiments show that all the species studied coordinate their limbs in the same way, so that the angle to which a particular segment of a limb can bend varies together with the angles that the other limb segments bend. This coordination implies that the movement of the center of body mass is regulated and energy is saved.

Along with providing new insight into how walking evolved, these findings may aid research into new approaches to treat walking impairments in humans and other animals.

DOI: https://doi.org/10.7554/eLife.38190.002

However, there is a gap in trying to relate the neural command signals to the mechanics of the COM. The COM is a virtual point in the trunk that shifts in space depending on the instantaneous configuration of the body. Its position is determined by the combined motion of the limb segments, as well as by trunk deformations. Therefore, in order to control the position of COM, the central pattern generators for locomotion must coordinate the motion of the limb segments (*Lacquaniti et al., 1999*; *Lacquaniti et al., 2002*).

Kinematic coordination of limb segments can be described by statistical methods such as principal component analysis (PCA), which projects movements onto a low-dimensional space thereby helping to detect invariant properties of coordination (*Daffertshofer et al., 2004*). Based on this approach, one law of inter-segmental coordination has been described in human locomotion, which involves the planar covariation of the temporal changes of the elevation angles of the lower limbs (*Borghese et al., 1996*). Specifically, limb segment rotations covary so that the three-dimensional (3D) trajectory of the elevation angles lies close to a plane.

The findings pertaining to the planar law are very reproducible (*Bianchi et al., 1998b*; *Bianchi et al., 1998a*; *Ivanenko et al., 2002*; *Ivanenko et al., 2007*; *Ivanenko et al., 2008*) and have been replicated in several laboratories (e.g. *Hicheur et al., 2006*; *Noble and Prentice, 2008*; *Barliya et al., 2009*; *Barliya et al., 2013*; *Hallemans and Aerts, 2009*; *Maclellan and McFadyen, 2010*; *Leurs et al., 2012*; *Wang et al., 2013*; *Aprigliano et al., 2016*). A planar law applies to both walking and running, as well as to other gait modes (*Grasso et al., 2000*; *Ivanenko et al., 2007*). It distinguishes between different developmental stages of human walk (*Cheron et al., 2001b*; *Cheron et al., 2001a*; *Ivanenko et al., 2004*; *Dominici et al., 2011*), as well as between normal and pathological gait (*Grasso et al., 2004b*; *Laroche et al., 2007*; *Leurs et al., 2012*; *Degelaen et al., 2013*; *Cappellini et al., 2016*; *Ishikawa et al., 2017*; *Wallard et al., 2018*). Importantly, it has been shown that the planar covariation in humans is not a trivial consequence of the geometrical and kinematic relationships between different limb segments (*Ivanenko et al., 2008*). Thus, newly walking toddlers lack an adult-like limb segment planar covariation, and children acquire it with walking experience (*Cheron et al., 2001b*; *Dominici et al., 2010*). In adults, the planar covariation can be violated in some conditions (e.g. when stooping and grasping an object during walking) or can

collapse in a simple linear relationship in other conditions (e.g. during stepping in place, *Ivanenko et al., 2008*). Also, spinal cord injured patients often lack the planar covariation (*Grasso et al., 2004a*).

The functional relevance of the synergic control of segmental motions lies in a reduction of the degrees of angular motion to two, thus matching the degrees of freedom of linear motion of the COM in the sagittal plane. In fact, a significant correlation has been found between the specific orientation of the plane, or equivalently the phase shift between the angular motion at the shank and foot, and the net mechanical power output at the COM at different speeds of walking (*Bianchi et al., 1998a*; *Bianchi et al., 1998b*). A consistent reduction of the degrees of angular motion has also been observed when additional segments are included in a principal component analysis of motion in the sagittal plane (*Mah et al., 1994*; *Borghese et al., 1996*; *Daffertshofer et al., 2004*; *Wang et al., 2013*; *Dewolf et al., 2018*).

How general across animal species is the planar law of intersegmental coordination? Does it happen to be a basic principle of terrestrial locomotion? One might expect this to be the case, given the potential connection linking the neuromuscular control patterns, the kinematic synergy, and energy saving at the COM. Initial evidence for the application of the planar law beyond the human species comes from the observation of a similar law in bipedal walking of Japanese macaques (*Ogihara et al., 2012*) and common quails (*Ogihara et al., 2014*), as well as in quadrupedal walking Rhesus monkeys (*Courtine et al., 2005*), dogs (*Catavitello et al., 2015*) and cats (*Shen and Poppele, 1995*; *Lacquaniti et al., 1999*).

However, apart from the above-mentioned studies, the planar law has received little attention so far in the context of comparative studies of animal locomotion. Our aim here was to look into the synergic control of segmental motions during terrestrial walking in a large set of mammals and birds. It is worth stressing that considering the large variety of body size, posture, limb configuration and segment proportions, one might not expect the same inter-segmental coordination across different animal species. Using a computational approach similar to that proposed by *Gatesy and Pollard (2011)*, *Figure 1A* illustrates the examples of permissible combinations of elevation angles assuming a fixed endpoint and constant hip (or shoulder) height during stance (Notice that the results of *Figure 1* do not change substantially if one relaxes the constraint of a constant hip (or shoulder) height during stance, since the vertical excursion of these proximal joints is relatively small during walking. Thus, the vertical oscillations due to the pendulum-like behavior of the limb during stance are 9.6±3.8% [mean±SD across all animals] of HL length for hip height, while the corresponding oscillations of shoulder height are 11.4±4.3% of FL length). Notice that the potential range of segment motions varies substantially across animals. For instance, the segment that can rotate the most is the foot for the mandrill while it is the shank for the avocet (see the relative size and shape of the ellipsoids in *Figure 1A*). Thus, the height of the hip (or shoulder) above ground and the limb segment proportions impose very different constraints on the range of angular motion across animals. Furthermore, the same planar covariation law may not apply to the forelimb (FL) angles and to the hindlimb (HL) angles of the same animal, nor may it apply to different animals. For instance, transfer of FL angles to HL angles in the cheetah results in aberrant trunk deformations, while transfer of HL angles of the camel to HL angles of the flamingo fails to predict a realistic hip height of the ipsilateral and contralateral limbs (*Figure 1B*).

We examined the basic kinematic patterns of limb motion in several animal species, belonging to diverse taxonomic groups of birds and therian mammals. Specifically, we studied 54 different animal species belonging to 2 classes and 18 orders, including 10 species of birds and 44 species of mammals, whose body mass spanned 5 orders of magnitude. Most recordings involved free walking in a natural environment.

## Results

### General gait parameters

The animals we analyzed varied widely in size and mass, from the mouse (typical mass 0.03 kg) to the elephant (about 4000 kg, *Table 1*). General gait parameters are reported in *Figure 2—figure supplement 2*. The stride duration ranged between 0.3 and 3 s, the longest in the hippopotamus (3 s), and the shortest in the mouse (0.3 s) (*Figure 2—figure supplement 2D*). The mean trunk

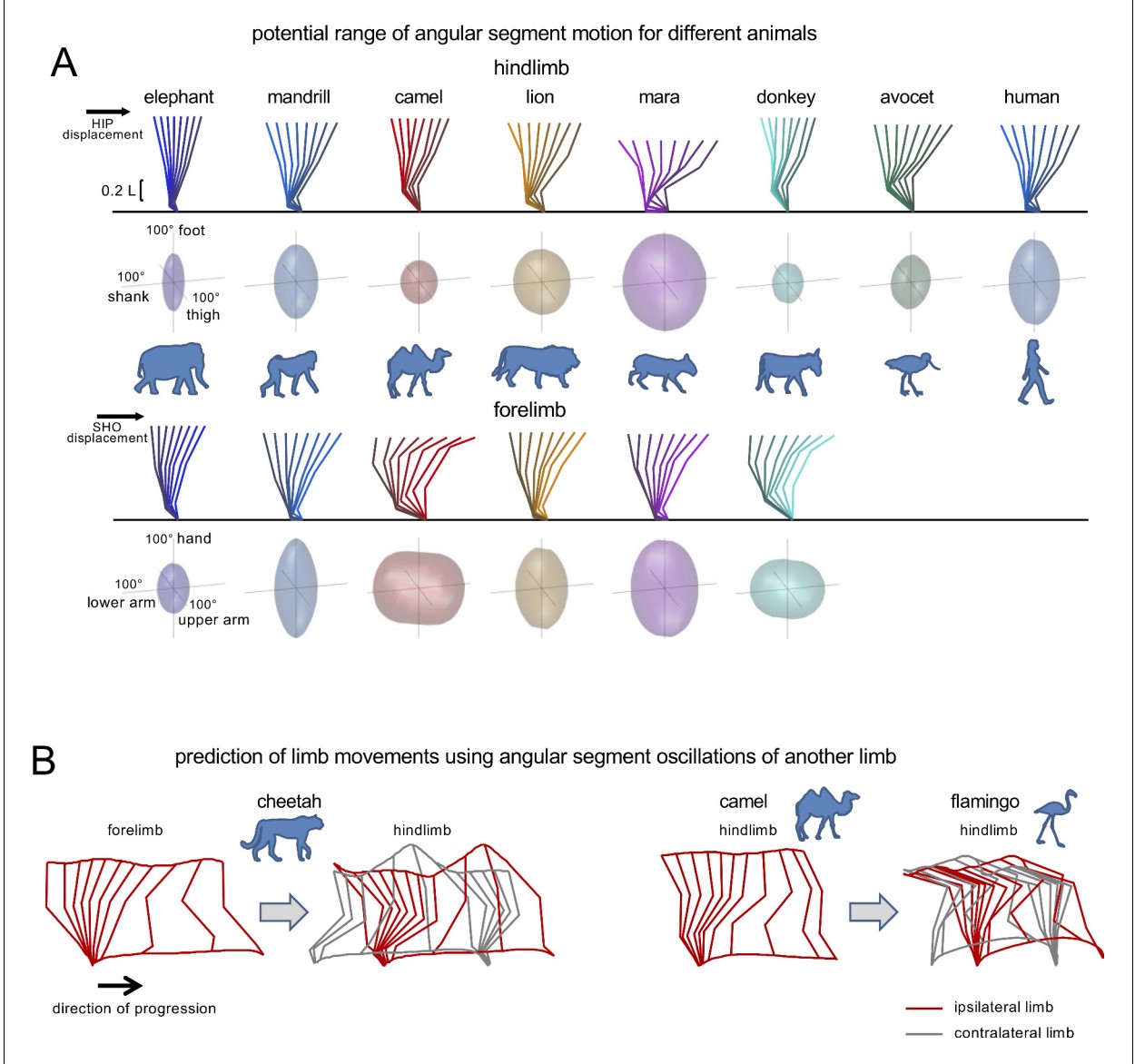

**Figure 1.** Interspecific comparison of angular movements during walking using a tri-segmented limb model. (**A**) Potential range of angular segment motion for representative animals of different orders during stance assuming a fixed endpoint and constant hip (or shoulder) height (using a similar computation as *Gatesy and Pollard, 2011*). Upper panels for hindlimb (HL) and forelimb (FL) show the examples of corresponding stick diagrams and lower panels illustrate ellipsoids reflecting permissible combinations of elevation angles derived from this computation. The height of the hip (HIP) and shoulder (SHO) for HL and FL, respectively, was defined as the average distance from the ground (calculated from our experimental data) for illustrated animals, expressed in limb length. (**B**) Artefacts of prediction of limb movements using angular segment oscillations of another limb and actual endpoint translation. The grey color refers to the stick diagram of the gait cycle of the contralateral limb. *Left panels:* Transfer of FL angles of cheetah to HL results in absurd trunk deformation due to unrealistic and not matched motion of the ipsilateral (red) and contralateral (grey) hips. *Right panels:* Transfer of HL angles of camel to HL of flamingo also fails to predict a realistic hip height of the ipsilateral and contralateral limbs. The data about the relative limb segment lengths and limb height for these animals are taken from the current study. Source files are available in the SourceData1-Figure1. zip file.

DOI: https://doi.org/10.7554/eLife.38190.003

The following source data is available for figure 1:

**Source data 1.** Interspecific comparison of angular movements during walking.

DOI: https://doi.org/10.7554/eLife.38190.004

orientation during walking corresponded to the most erect posture in humans (93° relative to the horizontal) and gibbons (76°), less vertical posture in birds (~10–45°), and nearly horizontal trunk in

**Table 1.** Analyzed animals.

The table shows the taxonomic classification with class, order and scientific name as reported in http://tolweb.org/tree/, http://www.arkive.org/, https://animaldiversity.org/, and the English name. Animals are sorted in the following order: birds - anseriformes, charadriiformes, ciconiiformes, columbiformes, galliformes, gruiformes, phoenicopteriformes, struthioniformes - and mammals - artiodactyla, carnivora, dasyuromorphia, didelphimorphia, hyracoidea, perissodactyla, primates, proboscidae, rodentia, scandentia. The locations where the animals were recorded are labeled by capital letters: F- Falconara zoo (Falconara, Italy); N - Iacchelli Farm (Nemi, Italy); R - Zoo of Rome (Rome, Italy); J - Nara City (Nara, Japan); LF – Fischer laboratory (*Fischer et al., 2002*); LA – Akay laboratory (*Akay et al., 2014*); LR – our laboratory (Rome, Italy). Next, we identify each species with a serial number and the first three letters of the corresponding taxonomic order, we report the number of analyzed animals, the typical body weight of adults (data from the literature), the mean speed for the recorded strides, the approximate dimensionless speed (Froude number, Fr), and the total number of analyzed strides for hindlimbs and forelimbs. For humans, we recorded both overground and treadmill walking (at 5 km/hr) in five subjects (data pooled together). For mice, video recordings were used from a previously published study (Movie S1 in *Akay et al., 2014*), while the kinematic data of the six mammals marked by asterisks are derived from *Fischer et al. (2002)*. In the latter case, the number of averaged strides varied: at least 15, but up to 300 step cycles were averaged.

| Order | Animals | Location | Label | N of animals | Typical weight (kg) | V (m/s) | Fr | N of strides HL | N of strides FL |
|---|---|---|---|---|---|---|---|---|---|
| **Birds** | | | | | | | | | |
| Anseriformes | Goose (*Alopochen aegyptiaca*) | F | 1Ans | 2 | 1.9 | 0.66 | 0.12 | 9 | - |
| Charadriiformes | Avocet (*Recurvirostra avosetta*) | F | 1Cha | 6 | 0.3 | 0.43 | 0.06 | 8 | - |
| Ciconiiformes | Ibis (*Threskiornis aethiopicus*) | F | 1Cic | 3 | 15 | 0.54 | 0.09 | 5 | - |
| Columbiformes | Pigeon (*Columba livia*) | N | 1Col | 3 | 0.4 | 0.72 | 0.32 | 7 | - |
| Galliformes | Guinea fowl (*Numida meleagris*) | N | 1Gal | 5 | 1.3 | 0.48 | 0.06 | 11 | - |
|  | Peacock (*Pavo cristatus*) | R | 2Gal | 4 | 4.4 | 0.60 | 0.06 | 11 | - |
| Gruiformes | Crane (*Balearica regulorum*) | F | 1Gru | 2 | 3.5 | 0.39 | 0.04 | 5 | - |
| Phoenicopteriformes | Flamingo (*Phoenicopterus roseus*) | F | 1Pho | 6 | 2.5 | 0.36 | 0.02 | 11 | - |
| Struthioniformes | Rhea (*Rhea americana*) | R | 1Str | 5 | 25 | 0.56 | 0.04 | 11 | - |
|  | Ostrich (*Struthio camelus*) | F | 2Str | 6 | 110 | 0.57 | 0.03 | 10 | - |
| **Mammals** | | | | | | | | | |
| Artiodactyla | Addax (*Addax nasomaculatus*) | R | 1Art | 1 | 93 | 1.07 | 0.12 | 2 | 2 |
|  | Ox (*Bos taurus*) | F | 2Art | 2 | 755 | 0.71 | 0.07 | 6 | 5 |
|  | Camel (*Camelus bactrianus*) | F | 3Art | 2 | 475 | 1.08 | 0.08 | 10 | 7 |
|  | Goat (*Capra hircus*) | F | 4Art | 2 | 45 | 1.17 | 0.16 | 3 | 2 |
|  | Waterbuck (*Kobus ellipsiprymnus*) | F | 5Art | 2 | 215 | 0.82 | 0.07 | 8 | 7 |
|  | Deer (*Dama dama*) | F | 6Art | 4 | 56 | 0.95 | 0.12 | 10 | 7 |
|  | Giraffe (*Giraffa camelopardalis*) | F,R | 7Art | 4 | 1560 | 1.21 | 0.06 | 9 | 10 |
|  | Hippopotamus (*Hippopotamus amphibious*) | F | 8Art | 1 | 2250 | 0.37 | 0.01 | 7 | 1 |
|  | Llama (*Lama glama*) | F | 9Art | 3 | 143 | 0.53 | 0.04 | 5 | 5 |
|  | Lechwe (*Kobus megaceros*) | R | 10Art | 4 | 175 | 0.79 | 0.07 | 7 | 5 |
|  | Nyala (*Tragelaphus angasii*) | F | 11Art | 3 | 91 | 0.57 | 0.03 | 5 | 4 |
|  | Oryx (*Oryx dammah*) | F | 12Art | 2 | 190 | 0.97 | 0.12 | 10 | 10 |
|  | Sika deer (*Cervus nippon*) | J | 13Art | 3 | 63 | 0.78 | 0.09 | 6 | 6 |

*Table 1 continued on next page*

*Table 1 continued*

| Order | Animals | Location | Label | N of animals | Typical weight (kg) | V (m/s) | Fr | N of strides HL | N of strides FL |
|---|---|---|---|---|---|---|---|---|---|
| Carnivora | Dog (Canis lupus familiaris) | R | 1Car | 6 | 35 | 1.06 | 0.24 | 35 | 36 |
| | Cat (Felis catus) | R | 2Car | 2 | 8 | 1.00 | 0.27 | 6 | 5 |
| | Cheetah (Acinonyx jubatus) | F | 3Car | 2 | 50 | 0.87 | 0.10 | 10 | 9 |
| | Lion (Panthera leo) | F | 4Car | 1 | 130 | 1.32 | 0.17 | 10 | 10 |
| | Lynx (Lynx lynx) | F | 5Car | 1 | 20 | 0.81 | 0.11 | 10 | 10 |
| | Wolf (Canis lupus) | F | 6Car | 1 | 40 | 1.05 | 0.20 | 6 | 6 |
| | Ocelot (Leopardus pardalis) | F | 7Car | 1 | 13 | 0.78 | 0.15 | 9 | 9 |
| | Cougar (Puma concolor) | F | 8Car | 2 | 62 | 1.05 | 0.18 | 10 | 10 |
| | Suricate (Suricata suricatta) | F | 9Car | 6 | 0.7 | 0.63 | 0.29 | 10 | 8 |
| | Tiger (Panthera tigris) | F | 10Car | 2 | 257 | 1.22 | 0.13 | 8 | 8 |
| Dasyuromorphia | Kowari (Dasyuroides byrnei;*) | LF | 1Das | 2 | 0.145 | * | * | * | * |
| Didelphimorphia | Short-tailed opossum (Monodelphis domestica*) | LF | 1Did | 2 | 0.092 | * | * | * | * |
| Hyracoidea | Rock hyrax(Procavia capensis*) | LF | 1Hyr | 2 | 1.2 | * | * | * | * |
| Perissodactyla | Donkey (Equus asinus) | F | 1Per | 2 | 203 | 0.81 | 0.07 | 16 | 12 |
| | Pony (Equus) | F | 2Per | 1 | 50 | 1.02 | 0.16 | 11 | 11 |
| | Tapir (Tapirus terrestris) | R | 3Per | 2 | 200 | 0.99 | 0.12 | 2 | 1 |
| | Zebra (Equus burchellii) | F | 4Per | 4 | 310 | 1.18 | 0.12 | 11 | 11 |
| Primates | Gibbon (Hylobates lar) | F | 1Pri | 3 | 5 | 1.03 | 0.23 | 6 | - |
| | Lemur catta (Lemur catta) | F,R | 2Pri | 7 | 3.9 | 0.66 | 0.14 | 10 | 3 |
| | Macaque (Macaca fuscata) | R | 3Pri | 2 | 9.8 | 0.86 | 0.14 | 2 | 0 |
| | Mandrill (Mandrillus sphinx) | R | 4Pri | 3 | 18 | 0.55 | 0.05 | 7 | 4 |
| | Spider monkey (Ateles fusciceps) | F | 5Pri | 3 | 9 | 0.75 | 0.13 | 3 | 2 |
| | Human (Homo sapiens) | LR | 6Pri | 6 | 68 | 1.46 | 0.27 | 67 | - |
| Proboscidea | Elephant (Elephas maximus) | R | 1Pro | 1 | 4050 | 1.24 | 0.07 | 6 | 6 |

*Table 1 continued*

| Order | Animals | Location | Label | N of animals | Typical weight (kg) | V (m/s) | Fr | N of strides HL | N of strides FL |
|---|---|---|---|---|---|---|---|---|---|
| Rodentia | Capybara (Hydrochoerus hydrochaeris) | F | 1Rod | 4 | 51 | 0.68 | 0.10 | 5 | 3 |
| | Cavy (Galea musteloides*) | LF | 2Rod | 2 | 0.3 | * | * | * | * |
| | Porcupine (Hystrix cristata) | F | 3Rod | 2 | 20 | 1.17 | 0.50 | 4 | 3 |
| | Mara (Dolichotis patagonum) | F | 4Rod | 3 | 8.1 | 0.82 | 0.18 | 9 | 9 |
| | Rat (Rattus norvegicus*) | LF | 5Rod | 3 | 0.35 | * | * | * | * |
| | Mouse (Mus) | LA | 6Rod | 1 | 0.03 | 0.52 | 0.34 | 3 | 3 |
| Scandentia | Common treeshrew (Tupaia glis*) | LF | 1Sca | 2 | 0.18 | * | * | * | * |

DOI: https://doi.org/10.7554/eLife.38190.005

quadrupeds (8 ± 7°) (*Figure 2—figure supplement 2E*). Animals were free to move spontaneously at their natural speed and each limb was on the ground for more than a half of the gait cycle (*Figure 2—figure supplement 2C*), as expected for walking gait. On average, the contacts of hindlimbs with the ground were shorter than those of forelimbs by ~10% of the gait cycle in quadrupeds (69 ± 5% for HL and 74 ± 4% for FL, n = 35, p<0.001, paired t-test), as Hildebrand (*Hildebrand, 1976*) also reported for some animals.

Touchdown events of the homologous limb pairs (i.e. left and right HL, or left and right FL) were almost equally spaced in time, around 50% of the gait cycle (between left and right limbs), so that all recorded gaits were roughly symmetrical (*Hildebrand, 1976*; *Cartmill et al., 2002*; *Abourachid, 2003*; *Frigon, 2017*). Most quadrupedal animals adopted the lateral gait pattern (i.e. when the HL touchdown is followed by the ipsilateral FL touchdown, $t_{FL} < t_{FLcontr}$, *Figure 2—figure supplement 2B*), consistent with the literature (*Miller et al., 1975*; *Abourachid, 2003*; *Schmitt, 2003*; *Righetti et al., 2015*), while primates showed the diagonal sequence (even though several primate species can also use a lateral sequence) (*Hildebrand, 1976*; *Cartmill et al., 2002*; *Frigon, 2017*). The gait of some quadrupeds can be defined as lateral sequence-diagonal couplets, since footfalls of contralateral limbs were almost synchronous (e.g. for the hippopotamus $t_{FL} = 41\%$, $t_{FLcontr} = 91\%$), so that their gait was almost diagonal even though with a lateral sequence.

In sum, a wide range of recorded animals demonstrated significant differences in the stride duration, trunk orientation, limb lengths and bilateral footfall patterns (*Figure 2—figure supplement 2*). There were also differences in the range of angular limb segment movements between animals and between HL and FL in quadrupeds, which will be reported in the following section.

## Amplitudes of limb segment oscillations in HL and FL

*Figure 2* illustrates the range of angular motion (ROM) of HL and FL segment elevation angles. For mammals, the ROMs of thigh, shank, and foot elevation angles were on average 38 ± 22°, 62 ± 16°, 71 ± 23°, respectively, and for the scapula, upper arm, lower arm and hand they were 29 ± 13°, 50 ± 22°, 75 ± 20°, 96 ± 33°, respectively. For birds, the ROM of the thigh segment was smaller than in mammals (on average by ~27°, p<0.00001, unpaired t-test), while it was larger for the shank (by ~16°, p=0.001) and it did not differ significantly for the foot (p=0.64) (*Figure 2*).

*Figure 2—figure supplement 2F* illustrates the ranges of linear limb movements. For all animals, horizontal limb endpoint excursions were significantly larger than the vertical ones, and for some animals they were relatively small (e.g. for artiodactyls) while for others (e.g. for carnivores, empty squares in *Figure 2—figure supplement 2F*) they exceeded the limb length (1L). In general, for quadrupeds, horizontal limb excursions were significantly greater for FL (0.95L) than for HL (0.87L). The animals walked at their natural speeds and some differences in the normalized limb endpoint excursions (*Figure 2—figure supplement 2F*) could be related to variations in the walking speed across animals. However, the reported difference between horizontal HL and FL endpoint excursions could not be related to walking speed since we compared them for the same animals. These inter-limb differences were expected, given the corresponding differences in the relative stance phase duration between the limbs (*Figure 2—figure supplement 2C*).

## Planar covariation of limb segment elevation angles

We used serially homologous HL and FL segments and models for comparing the kinematics of the HL and FL, starting from the distal segment: foot-hand, shank-lower arm, and thigh-upper arm. However, the scapula segment also undergoes significant rotations in the sagittal plane (*Figure 2*). While PCA can be applied also in four dimensions for FL, using a tri-segmental model makes it easier to compare the kinematic synergies between FL and HL. Therefore, for FL we used two separate tri-segmental models (*Fischer and Blickhan, 2006*): $FL_{low}$ – 'upper arm–lower arm–hand' and $FL_{upp}$ – 'scapula–upper arm–lower arm' (*Figure 3A* right panel).

We found that the planar covariation law of limb segment motions holds for walking in all recorded animal species, despite significant differences in body size, limb segment configuration and gait parameters (*Figure 2* and *Figure 2—figure supplement 2*). *Figure 3* shows the results of PCA applied to the kinematic data of different animals.

*Figure 3A* shows examples of the ensemble-averaged elevation angles (across strides) as a function of the normalized gait cycle (upper panels) and a corresponding three-dimensional view of these

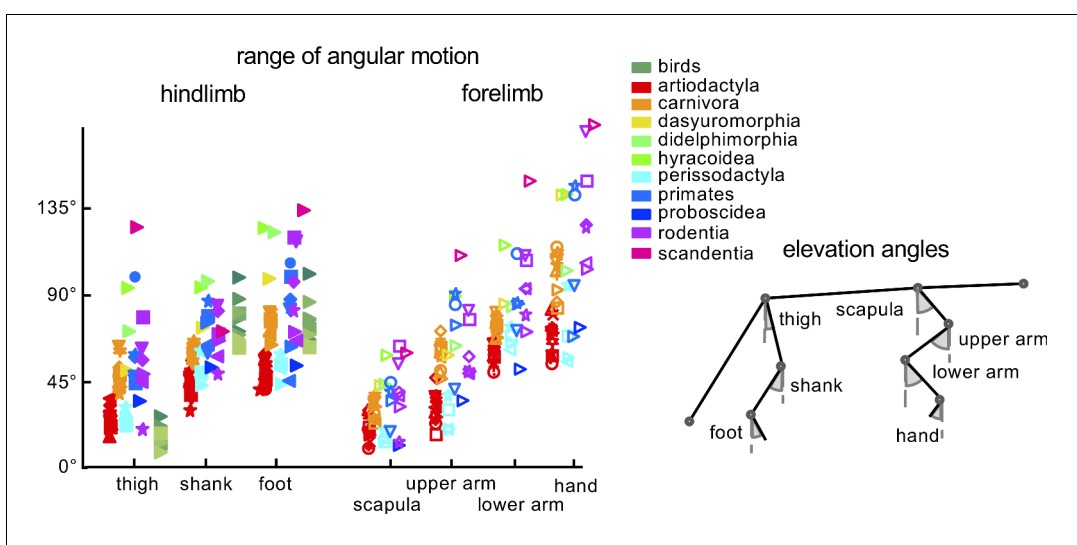

**Figure 2.** General kinematic characteristics. Range of angular motion (ROM) of limb segment elevation angles (see schematic drawing on the right) averaged across strides for each animal. Different markers and colors refer to different taxonomic orders. Empty and filled markers refer to FL and HL values, respectively. Source files are available in the SourceData2-Figure2.zip file.

DOI: https://doi.org/10.7554/eLife.38190.006

The following source data and figure supplements are available for figure 2:

**Source data 1.** Range of angular motion of segments elevation angles.
DOI: https://doi.org/10.7554/eLife.38190.010
**Figure supplement 1.** Anatomical landmarks and segments used for the kinematic model of bipedal.
DOI: https://doi.org/10.7554/eLife.38190.007
**Figure supplement 2.** General gait parameters.
DOI: https://doi.org/10.7554/eLife.38190.008
**Figure supplement 2—source data 1.** General gait parameters.
DOI: https://doi.org/10.7554/eLife.38190.009

angles (lower panels) for one bipedal (avocet) and one quadrupedal (elephant) animal. The foot at HL and the hand at FL touchdown corresponds to the top of the loops in the lower panels. The trajectories progress in the counter clockwise direction of the loops. The grids represent the best fitting planes, defined by the first two eigenvectors of the PCA. Note the differences in the orientation of the covariation plane at HL between the two animals, and between HL and FL for the elephant. Note also a roughly similar orientation of the covariation plane between the two models of FL ($FL_{upp}$ and $FL_{low}$, *Figure 3A*).

Planarity of the data was quantified by computing the percentage of variance accounted for by the third eigenvector ($PV_3$) of the data covariation matrix: the closer is $PV_3$ to 0, the smaller the deviation from planarity. The results showed that the planarity was obeyed by all species ($PV_3$ ranged from 0.04% to 5.3% across all limbs and animals) (*Figure 3B*).

We examined mainly the intersegmental coordination of the elevation angles, rather than that of the relative joint angles (so called anatomical angles), because the former capture the overall limb configuration in external space. Indeed, the elevation angles identify the orientation of each segment relative to the direction of gravity (vertical direction). Moreover, the time course of the anatomical angles of flexion-extension in human locomotion is more variable across subjects and trials than that of the elevation angles, and the planarity of the anatomical angles trajectories is weaker (*Borghese et al., 1996*; *Barliya et al., 2009*). In our recorded animals, when we applied the PCA to the anatomical angles (hip, knee and ankle for HL, and shoulder, elbow and wrist for FL), the planarity indexes ($PV_3$) were: $2.7 \pm 2.3\%$ for HL angles (ranging from 0.02% in guinea fowl to 8.0% in porcupine) and $4.0 \pm 2.5\%$ for FL angles (ranging from 0.7% in ox to 13.9% in porcupine). Therefore, although also the anatomical angles trajectories tend to be constrained close to one plane, the

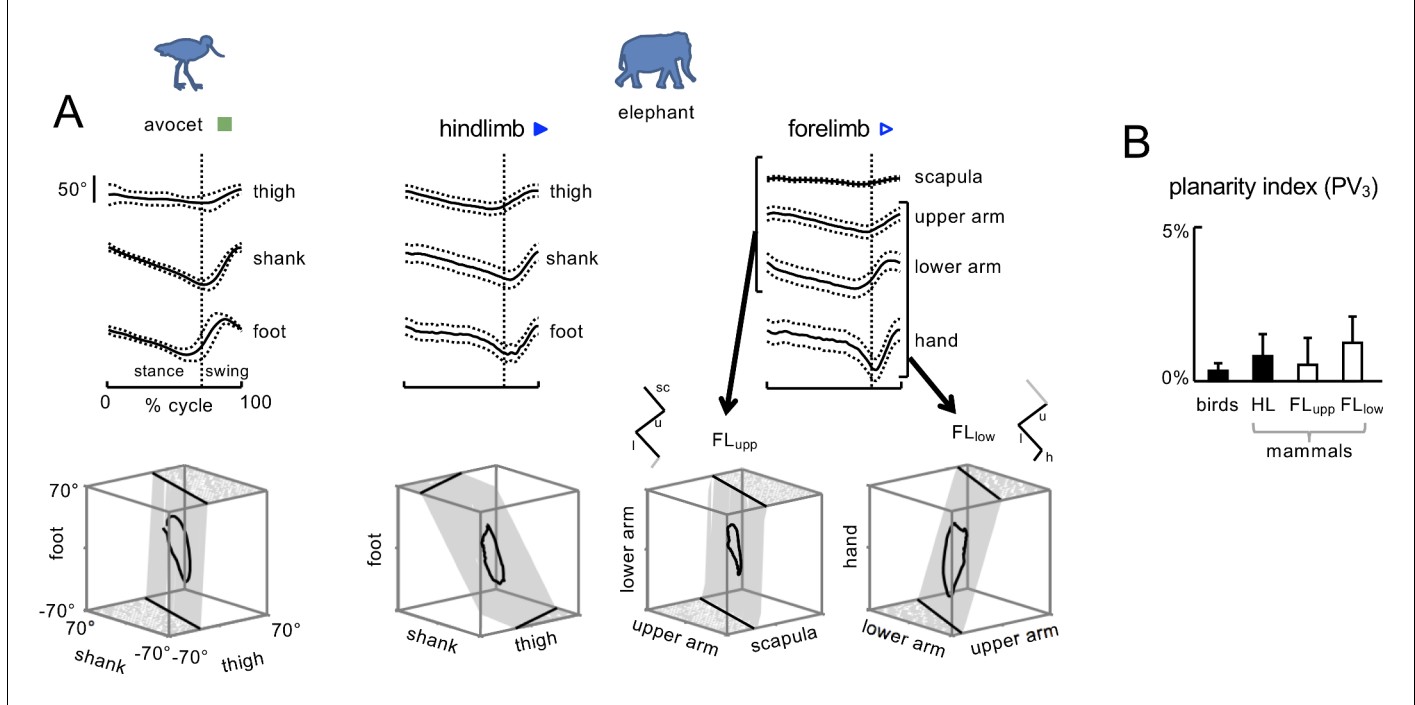

**Figure 3.** Planar covariation of limb segment angular motion. (A) Ensemble-averaged (±SD across strides) limb segment elevation angles plotted vs. normalized gait cycle and three-dimensional gait loops with corresponding interpolation planes on the bottom are shown for one bird (avocet) and one mammal (elephant). For FL, planar covariation was examined using different tri-segmental FL models: $FL_{upp}$ – 'scapula–upper arm–lower arm' and $FL_{low}$ – 'upper arm–lower arm–hand'. Note roughly similar orientation of the covariation plane between these two models of FL and different orientation compared to the HL model. (B) Planarity index expressed in percent of variance (PV) accounted for by the third principal component (PC) for HL and FL of animals ($PV_3 = 0$ for ideal planarity). Source files are available in the SourceData3-Figure3.zip file.

DOI: https://doi.org/10.7554/eLife.38190.011

The following source data and figure supplements are available for figure 3:

**Source data 1.** Planar covariation of limb segment motion.
DOI: https://doi.org/10.7554/eLife.38190.016

**Figure supplement 1.** Direction cosines of the normal to the covariation plane (the dot product of $u_3$ with the unit vector along each of the three axes: $u_{3t}$, $u_{3s}$, $u_{3f}$ for HL and $u_{3sc}$, $u_{3u}$, $u_{3l}$ and $u_{3u}$, $u_{3l}$, $u_{3h}$ for $FL_{upp}$ and $FL_{low}$, respectively) for all animals.
DOI: https://doi.org/10.7554/eLife.38190.012

**Figure supplement 1—source data 1.** Direction cosine of the normal to the covariation plane.
DOI: https://doi.org/10.7554/eLife.38190.013

**Figure supplement 2.** Gait loops in birds and mammals.
DOI: https://doi.org/10.7554/eLife.38190.014

**Figure supplement 2—source data 1.** Gait loops.
DOI: https://doi.org/10.7554/eLife.38190.015

planar coordination of the elevation angles is stronger and less variable, in agreement with what was previously reported for human walking (*Borghese et al., 1996*; *Barliya et al., 2009*).

It is interesting to note that the shape of the 3D trajectories generated by the three elevation angular waveforms differed across animals and limbs (*Figure 3—figure supplement 2*). All trajectories presented a closed loop, as expected by the cyclic nature of the gait, and were elongated in the direction of $PC_1$ given that $PC_1$ explains the largest fraction of variance (*Figure 3—figure supplement 2A*). We quantified the relative width of the loop by computing the relative amplitude of $PC_2$ with respect to $PC_1$ (*Figure 3—figure supplement 2B*). The width of the loop in birds (0.54 ± 0.11) was significantly larger than for HL in mammals (0.26 ± 0.06) (p=0.00002, unpaired t-test). In addition, there was a difference between HL and $FL_{low}$ (0.49 ± 0.11) and between HL and $FL_{upp}$ (0.31 ± 0.08) gait loops in quadrupeds (p<0.005, paired t-test). There were no significant differences between the loops of birds and $FL_{low}$ of mammals (p=0.25).

## Orientation of the covariation plane of limb segment motion

While planarity of the 3D loops held for all animals (*Figure 3B*), the orientation of the covariation plane differed, due to different phase relationships between elevation angles. The third eigenvector ($u_3$) of the covariance matrix is orthogonal to the best fitting plane and characterizes its orientation. *Figure 3—figure supplement 1* illustrates the direction cosines of the normal to the plane (i.e. the dot products of $u_3$ with the unit vectors along each of the three axes) for all animals. One can notice a greater scatter of the $u_3$ components of HL across mammals than the corresponding components in birds or those of FL in mammals.

One way to visualize the $u_3$ vector in a 3D space for all animals is to plot it on a sphere as shown in *Figure 4*. Note a higher $u_3$ dispersion for HL (*Figure 4A*) than for FL (*Figure 4B*), consistent with the greater variability of $u_3$ components in HL noticed above. Using the empirical shape criterion (*Fisher et al., 1993*), we distinguished the girdle distribution (a type of distribution of directions with a concentration about a given plane) of the eigenvectors in mammals HL from the clustered distribution of birds and of mammals FL. In particular, this criterion revealed that the four groups (birds HL, mammals HL, mammals $FL_{upp}$, mammals $FL_{low}$) had concentration parameters significantly different (p<0.0001) and belonged to different distribution, respectively a von-Mises Fisher distribution with concentration parameters $k = 106.9$, a Kent distribution (girdle-like shape) with $k = 5.7$, and two other von Mises distribution with $k = 20.6$, $k = 33.7$. High values of the concentration parameters showed that the population mean directions were different across the four groups (p<0.0001). Furthermore, it is worth noting that the vectors were not dispersed randomly on a sphere, but tended to lay on the plane defined by the mean $u_1$ vector across animals (since $u_1$ was roughly the same; overall, it deviated from the mean $u_1$ by 2.4° [spherical standard error] for HL, by 2.7° for $FL_{upp}$ and by 2.6° for $FL_{low}$). The lower panels of each box in *Figure 4A and B* illustrate the rotation of the $u_3$ vectors on this plane for each animal and limb. This plane is perpendicular to the averaged $u_1$ vector across animals, and $\alpha$ defines the angle of rotation of each $u_3$ on the plane.

In sum, the full limb behavior in all walking animals can be expressed as two principal components identifying a given covariation plane (*Figure 3*). While the orientation of the covariation plane of the FL appears fairly conserved across species (*Figure 4B*), the covariation plane of HL varies across mammalian species by a rotation ($\alpha$-angle) about a well-defined axis (*Figure 4A*).

We searched for the presence of a phylogenetic signal in the wide scatter of $\alpha$-values of rotation for HL across animal species, in order to frame the data scatter in an evolutionary context. Although the K index (*Blomberg et al., 2003*, see Materials and methods) we used for the presence of a phylogenetic signal in the $\alpha$-angle for HL was statistically significant, its value was rather low (K = 0.10, n = 54, p=0.04) (*Figure 4—figure supplement 1*), suggesting that the pattern of $\alpha$-angles distribution is hardly dependent on phylogenetic relatedness of the species considered. Such a pattern may occur when close relatives are less similar than distant ones.

## Relationship between limb parameters and rotation of the covariation plane

To search for biomechanical correlates of inter-species differences in the orientation of the covariation plane, we performed a linear regression between the value of $\alpha$-angle (*Figure 4A,B*) and the ROM of limb segment elevation angles, the phase shift between elevation angles (i.e. timing of their minima), and the ratio between limb segment lengths. The rationale for using these biomechanical parameters was that the animals differed significantly in terms of limb proportions (*Figure 2—figure supplement 1*), ROMs (*Figure 2*), and temporal sequence of minima of elevation angles (*Figure 5A*, bottom panels). In particular, the phase shifts between elevation angles waveforms are strongly related to the rotation of the covariation plane (*Bianchi et al., 1998b*; *Barliya et al., 2009*), and they can be assessed using the relative timing of the minima (*Bianchi et al., 1998b*; *Catavitello et al., 2015*).

We found that the values of $\alpha$-angle were best correlated with the phase shift $\Delta_{foot-shank}$ ($r^2 = 0.5$, p<0.0000001, *Figure 5A*). Other relatively high ($r^2 \geq 0.3$) significant correlations were: for HL, the ratios $L_{shank}/L_{foot}$ and $L_{thigh}/L_{foot}$ ($r^2 = 0.45$ and $r^2 = 0.31$, respectively, p<0.00004) and, for $FL_{upp}$, ROMs of the upper and lower arms ($r^2 = 0.40$ and $r^2 = 0.32$, respectively, p<0.0002), and the phase shift $\Delta_{upper\ arm-scapula}$ ($r^2 = 0.3$, p<0.0003, *Figure 5A* right panel). After controlling for a potential phylogenetic signal in the response (and, hence, non-independence of the residuals, see Materials

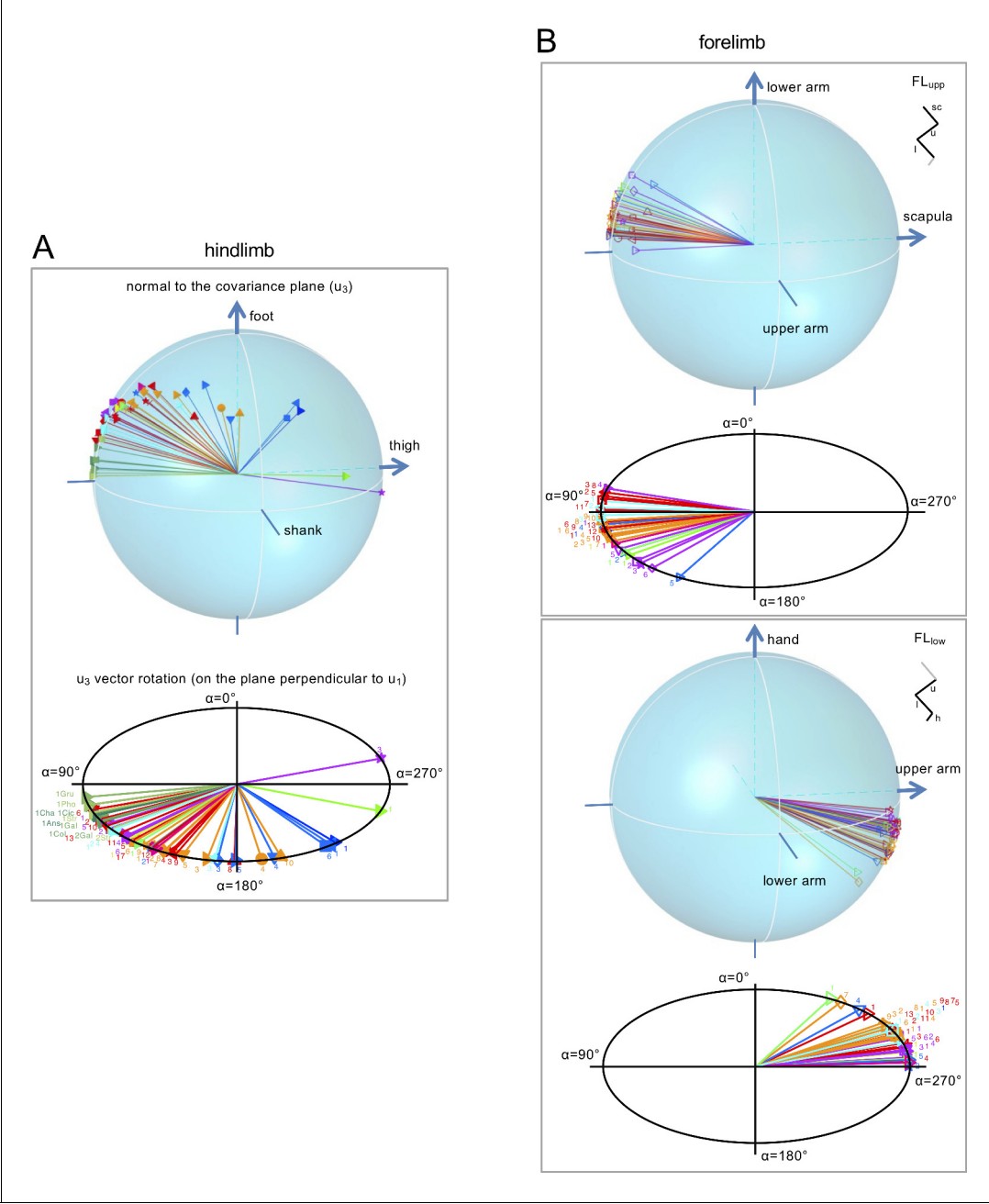

**Figure 4.** Orientation ($u_3$ vector) of the covariation plane for the HL (A) and FL (B) segment elevation angles. For FL, the normal to the plane for both $FL_{upp}$ and $FL_{low}$ tri-segmental models are shown. Top panels: spatial distribution of the normal ($u_3$) to the covariation plane in all animals. Each vector $u_3$ yields one point corresponding to the projection of the plane normal onto the unit sphere, the axes of which are the direction cosines with the axes of segment elevation angles ($u_{3t}$, $u_{3s}$, $u_{3f}$). Positive direction of the semi-axes is indicated by arrow. Lower panels: $u_3$ vectors for different animals lay on the plane (perpendicular to the averaged $u_1$ vector across animals), $\alpha$ - angle of rotation of $u_3$ on this plane ($\alpha$ = 0 corresponds to the projection of the shank, upper arm and lower arm semi-axes on that plane for HL, $FL_{upp}$ and $FL_{low}$, respectively). Animals are labeled as in *Table 1* and different colors and markers refer to different taxonomic orders and species, respectively. Note a higher data dispersion for HL compared to FL across animals. Source files are available in the SourceData4-Figure4.zip file.

DOI: https://doi.org/10.7554/eLife.38190.017

The following source data and figure supplements are available for figure 4:

**Source data 1.** Orientation of the covariation plane.

DOI: https://doi.org/10.7554/eLife.38190.020

**Figure supplement 1.** Reconstructed ancestral trait values of $\alpha$ for HL on a phylogeny.

*Figure 4 continued on next page*

*Figure 4 continued*

DOI: https://doi.org/10.7554/eLife.38190.018

**Figure supplement 1—source data 1.** Reconstructed ancestral trait values.

DOI: https://doi.org/10.7554/eLife.38190.019

and methods), we found that the $\alpha$-angle remained significantly correlated with $\Delta_{foot-shank}$ ($r^2 = 0.37$, p<0.00001) and $\Delta_{upper\ arm-scapula}$ ($r^2 = 0.31$, p<0.00001). For $FL_{low}$, all correlations were very weak ($r^2$ ~0.01–0.14), and involved much smaller rotations of the covariation plane (*Figure 4B*). There were also differences in the temporal sequence of minima of elevation angles between the limbs (*Figure 5B*). Even though the timing of minima for all segments occurred roughly around the stance-to-swing transition (since the relative stance duration was about 70% cycle, *Figure 2—figure supplement 2C*), the sequence of minima differed for HL and FL. For instance, one can notice that the distal segment (hand) of FL was the last to initiate the swing phase in contrast to the distal segment (foot) of HL (*Figure 5B*).

In *Figures 4* and *5*, we reported the parameters of the inter-segmental coordination in all animal species. To obtain a general template of HL and FL angular motions for each animal species, we averaged limb segment elevation angles across strides. However, some inter-stride variability in the orientation of the covariation plane and timing of the minima of elevation angles exists. Also, there were some limitations of our measurements (e.g. due to some variability in the walking speed across strides). Nevertheless, it is unlikely that the key differences across limbs and species can be accounted for or masked by inter-stride variability. We quantified the inter-stride variability in the animal species in which we recorded more than 15 strides, namely: dog, donkey and human (*Table 1*). First, the correlation between the averaged limb segment elevation angles and those of individual strides was high (on average r = 0.98, range 0.7-1, the data for all segments and animals being pooled together), consistent with repeatable kinematic data across steps in animals (*Faber et al., 2002*; *Kim et al., 2008*). Second, the sequence of timing of minima of elevation angles showed systematic features as well. For instance, the distal segment (hand) of FL always followed the lower arm segment ($\Delta_{hand-lower\ arm}$ being positive in all strides), in contrast to the distal (foot) segment of HL ($\Delta_{foot-shank}$ varied but was generally negative). Finally, the inter-stride and inter-individual variability in the orientation of the covariation plane ($u_3$ vector) was relatively small in comparison with the differences across animals (*Figure 4*). For instance, the angular standard deviation of the parameter $\alpha$ (the angle of rotation of $u_3$) across strides was 11° in dogs, 12° in donkeys and 6° in humans (the strides of all animals for each animal species being pooled together), and across animals it was 6° in dogs (n = 6), and 5° in humans (n = 6), while the differences in $\alpha$ across animal species were much larger (~180°, *Figure 4*).

## Discussion

We examined limb kinematics during terrestrial locomotion of 54 different animal species. The results showed that, despite significant variations in body size, mass, limb configuration (*Figure 2—figure supplement 1*), stride duration (*Figure 2—figure supplement 2D*) and relative amplitude of angular movements (*Figure 2*), the planar law of inter-segmental coordination held in all studied mammals and birds (*Figure 3—figure supplement 1* and *Figure 4*), suggesting that this kinematic synergy is ubiquitous in terrestrial locomotion.

### Inter-segmental coordination at hindlimbs and forelimbs

During forward progression, HL and FL segments oscillate back and forth with specific phasing relative to the footfall pattern (*Figure 3A*). We confirmed the validity of the planar covariation previously reported in humans (*Bianchi et al., 1998b*), macaques (*Courtine et al., 2005*; *Ogihara et al., 2012*), birds (quails, *Ogihara et al., 2014*), and dogs (*Catavitello et al., 2015*), and extended it to a large set of other animal species. Even though different recording systems were used in these previous studies, the planarity index ($PV_3$ ~1–3%) and the orientation of the covariation plane ($u_3$ vector) for the bird (*Ogihara et al., 2014*), human (*Bianchi et al., 1998a*) and dog (*Catavitello et al., 2015*) were similar to those reported in the current study (*Figure 3*), confirming the reliability of our

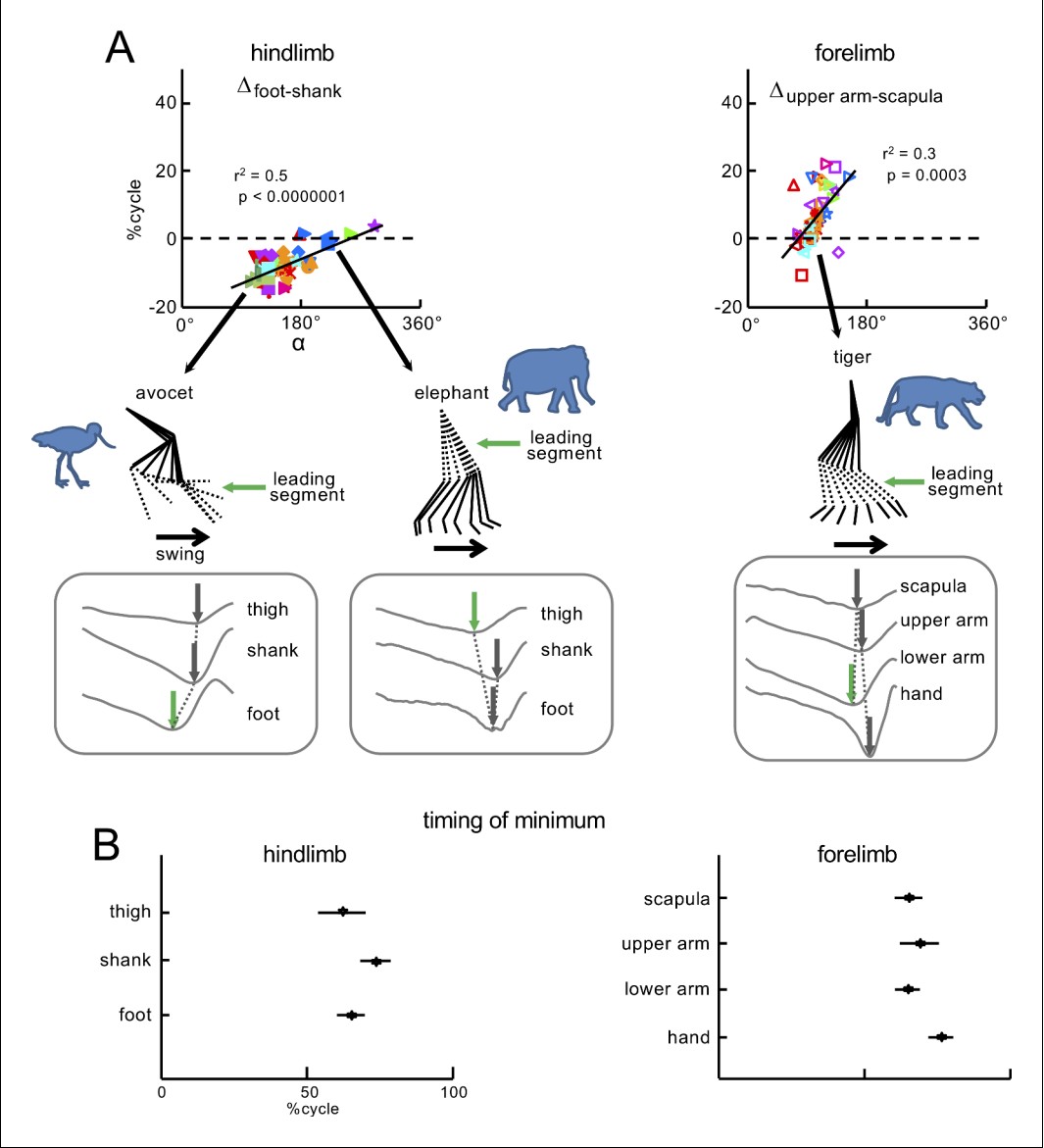

**Figure 5.** Significant correlations between the orientation of the covariation plane and the phase shifts between elevation angles. (**A**) Relationships between $\Delta_{\text{foot-shank}}$, $\Delta_{\text{upper arm-scapula}}$ (the differences between the timing of minima of elevation angles) and orientation of the $u_3$ vector ($\alpha$ angle in **Figure 4**). Linear regressions are also displayed with corresponding $r^2$ and p-values. On the bottom of panel A: examples of the limb segment elevation angles for HL of avocet and elephant, and FL of tiger. The sequence of minima is indicated by arrows. The leading segment, which corresponds to the first minimum and initiates the swing phase (green arrow) is highlighted by dotted lines in the stick diagrams. (**B**) Timing of minima (±SD) of the limb segment elevation angles of HL and FL of all animals expressed in percent of gait cycle. Source files are available in the SourceData5-Figure5.zip file.
DOI: https://doi.org/10.7554/eLife.38190.021

The following source data is available for figure 5:

**Source data 1.** Significant correlations between the orientation of the covariation plane and the phase shifts.
DOI: https://doi.org/10.7554/eLife.38190.022

kinematic recordings and suggesting that each animal adopted its own pattern of the inter-segmental coordination.

Since we recorded animals walking at their preferred speeds, some variability in the covariation plane orientation may be caused by variations in speed across strides. However, it is unlikely the

relative invariance in the orientation at the forelimbs (which in theory might parallel changes in speed) and in the sequence of timing of minima of elevation angles suggests that the key differences across the limbs and species is accounted for or masked by inter-stride variability, consistent with repeatable kinematic data across steps in animals (*Faber et al., 2002*; *Kim et al., 2008*). Also, we verified the inter-individual variability in a few species (dogs and humans, *Table 1*), and the angular standard deviation in the covariation plane orientation (~5–6°) was much smaller than the differences in $\alpha$ (the angle of rotation of $u_3$) across all animal species (~180°, *Figure 4*), suggesting that the data presented in *Figure 4* are representative for each animal species. One should also consider that the effect of speed may be animal- or gait-dependent. For instance, in human bipedal walking there is some rotation (although relatively small) of the covariation plane with speed (*Bianchi et al., 1998a*), while in human crawling the orientation of the covariation plane does not depend on speed (*MacLellan et al., 2017*). To obtain a general template of limb segment motion of animals walking at their preferred speeds, we analyzed averaged angles (as in other previous studies, e.g., *Fischer et al., 2002*). Further studies may reveal a nuanced dependence of planar covariation parameters on walking speed or gait in different animals.

Specific limb proportions may play an essential role in the kinematics and energetics of walking (*Leurs et al., 2011*), and they have an impact on a locomotor body schema used for controlling step length (*Ivanenko et al., 2011*). For instance, in humans elongation of the shank segment relative to the thigh segment (either surgically or using specially designed stilts) affects the amplitude of distal vs. proximal segment oscillations (*Dominici et al., 2009*) and the optimal step length and walking speed (*Leurs et al., 2011*). In line of principle, species with different limb proportions are free to employ identical angular movements, but there are biomechanical constraints of the articulated chains on the end effector positions (*Gatesy and Pollard, 2011*) and on the possibility of transferring angular changes between disproportionate limbs (*Figure 1B*, right panel). Limb segment proportions varied significantly across species (*Figure 2—figure supplement 1*). For instance, the $L_{shank}$/$L_{foot}$ and $L_{thigh}$/$L_{foot}$ ratios showed 10-fold differences in our sample of animals (range 0.5–5). Therefore, it is important to stress that the planar covariation holds for animals with very different limb segment configurations.

While biomechanics contributes to the planar law of inter-segmental coordination (e.g. we found that the $u_3$ vector rotation correlated with $L_{shank}$/$L_{foot}$ and $L_{thigh}$/$L_{foot}$), the orientation of the covariation plane reflects specific phase relationships in the control of segment motions (*Bianchi et al., 1998b*; *Lacquaniti et al., 2002*; *Ivanenko et al., 2008*). For instance, birds form a group of animals with a compact orientation of the $u_3$ vector close to the thigh axis (*Figure 4A*, see also *Ogihara et al., 2014*) while primates show rather variable $u_3$ orientation (*Figure 3—figure supplement 1*). Also, birds show characteristically wide gait loops, while for other animals the loops are much narrower for HL (*Figure 3—figure supplement 2*). Finally, for quadrupedal animals, the orientation of the covariation plane is noticeably different for HL and FL (*Figures 3* and *4*).

The latter finding represents a particularly interesting phenomenon that may shed further light on the functional difference between the limbs and their control. HL and FL kinematics are characterized by limb-specific differences in the orientation of the covariation plane (*Figures 3* and *4*), the width of the gait loop (the $FL_{low}$ loop was wider than the HL loop, *Figure 3—figure supplement 2*) and the amplitude (*Figure 2*) and phase (*Figure 5B*) of angular motion of distal segments. These results confirm previous observations about different orientation of the HL and FL covariation planes in Rhesus monkey (*Courtine et al., 2005*), dog (*Catavitello et al., 2015*) and human crawling (*MacLellan et al., 2017*), and point to the differential control of FL and HL segments in a wide range of mammals. The distinctive orientation of FL and HL segments (the elbow is facing posteriorly and the knee joint anteriorly), a stronger push-off function of HL (e.g. during jumping), and the differences in the leading segment (as assessed by temporal sequence of minima in the elevation angles around the stance-to-swing transition, *Figure 5*) may impose specific phase-relationships of FL and HL segment oscillations. Finally, neurophysiological differences in the neurotransmitter systems of FL versus HL spinal locomotor controllers (*Gerasimenko et al., 2009*), a strong asymmetry of projections from spinal controllers on neurons for FL versus HL areas of the motor cortex (*Zelenin et al., 2011*), and limb-specific features in the organization and coupling between FL and HL spinal controllers (*Shik and Orlovsky, 1965*; *Miller et al., 1975*; *Yamaguchi, 1986*) point to limb-specific organization of central pattern generators, with propriospinal linkages facilitating the coordination between FL and HL (*Frigon, 2017*). The temporal structure of FL and HL muscle activation patterns

is limb-specific too, as is the orientation of the covariation plane (e.g. in the dog, *Catavitello et al., 2015*).

Furthermore, the orientation of the covariation plane (normal to the plane, $u_3$) varies more for HL than for FL across animals (*Figure 4*). This finding is also compatible with larger kinematic changes in HL movements reported in previous studies. For instance, *Fischer et al. (2002)* reported variable lift-off configuration of HL with respect to FL, when comparing different gaits of the same animal. Thus, main kinematic adaptations seem to occur in hindlimbs rather than in forelimbs both across animals (*Figures 4* and *5*) and across gaits (*Fischer et al., 2002*). This implies considerable adaptability or flexibility in the phase relationships of HL segment motion, which will be considered in the following section.

We also searched for the presence of a phylogenetic signal in the scatter of $\alpha$-values of rotation for HL across animal species, and we found only a weak one. Indeed, the $\alpha$-values of close relatives were not more similar between each other than to the values of distant ones (*Figure 4—figure supplement 1*). This suggests that the planar covariation is a feature that has arisen independently several times during evolutionary history (*Ogihara et al., 2014*). A convergent evolution of this kinematic synergy may be due to both adaptation and constraints acting similarly in distantly related species. Adaptation would arise due to the advantage of a kinematic control law lying at the interface between neural commands and environment. Constraints would depend on the inherent biomechanical coupling between different limb segments.

## Phase relationships and functional interpretation of principal components

The planar covariation law may emerge from the coupling of neural oscillators with limb mechanical oscillators (*Lacquaniti et al., 2002*; *Lacquaniti et al., 2012*), by adjusting the phase of unit burst generators for each joint, segment or groups of muscles (*Grillner, 1981*; *Kiehn, 2016*). The basic mechanism of rhythmic movements is a phase control of muscle activity. In particular, myoelectric signal analysis demonstrated a burst-like temporal organization of basic muscle activation patterns shared by many animal species (*Giszter et al., 2010*; *Dominici et al., 2011*; *Lacquaniti et al., 2013*), consistent with the existence of a rhythm-generating layer or 'time-keeping function' of the central pattern generator for locomotion (*Prentice et al., 1995*; *McCrea and Rybak, 2008*). Because the activation patterns are pulsatile, muscle activations intervene only during limited time epochs at specific phases of the gait cycle to re-excite the intrinsic oscillations of the system when energy is lost (*Lacquaniti et al., 1999*; *Lacquaniti et al., 2012*). This represents a fundamental energy-saving principle of control.

The dynamic behavior of the musculo-skeletal system can be modeled through a linear combination of these basic muscle patterns, activated sequentially at touch-down, body-weight support, limb lift-off, and swing (*Lacquaniti et al., 2012*). The specific orientation of the planar covariation is related to the timing of basic muscle activation patterns. In humans, changes in the orientation of the covariation plane with walking speed (*Bianchi et al., 1998b*) or across different gaits (*Ivanenko et al., 2007*) are associated with changes in the timing of basic muscle activation patterns (*Ivanenko et al., 2004*; *Cappellini et al., 2006*). In dogs, the phase-coupling between the elevation angles differs systematically between HL and FL (*Figure 4*), just as the phase-coupling of the muscle activation patterns (*Catavitello et al., 2015*). Thus, although it is often assumed that central pattern generators control patterns of muscle activity, an equally plausible hypothesis is that they control patterns of limb segment motion (*Lacquaniti et al., 1999*; *Lacquaniti et al., 2002*), since the phase relationships between them are inherently inter-related.

The full limb behavior can be expressed as the two degrees-of-freedom planar motion for each animal, plus the rotation of the plane about a defined axis (*Figure 4*, see also *Ivanenko et al., 2007*). An analytical formulation of the law of inter-segmental coordination in human walking was introduced by *Barliya et al. (2009)*, using a mathematical model that represents the rotations of the elevation angles in terms of simple harmonic oscillators with appropriate phase shifts between them. This model can be generalized to the locomotion of other animal species. We found the highest correlation between the rotation of the covariation plane ($\alpha$-angle) and $\Delta_{\text{foot-shank}}$ (*Figure 5*). Therefore, the phase shift between foot and shank segments represents an important parametric tuning of the covariation plane rotation to adapt to animal-specific locomotor patterns (*Figure 4A*), walking speed

(*Bianchi et al., 1998b*), gait (*Ivanenko et al., 2007*) or walking on different support surfaces (*Dominici et al., 2010*).

*Figure 6* provides schematically the conceptual framework for modelling the foot-shank phase-shift, while approximating the three segment elevation angles with sinusoidal waveforms *Barliya et al. (2009)*. Notice that, as predicted, changing the foot-shank phase results in a progressive rotation of the planar covariation. Critically, the rotated planes (upper panels in *Figure 6A*) closely resemble the experimental planes of different animals (lower panels). Thus, by changing the phase of the foot segment waveform relative to the shank segment from 20° to −50° (corresponding to the same range of $\Delta_{\text{foot-shank}}$ in *Figure 5A*), the covariation plane rotates similarly to the plane rotation actually observed across animals (*Figure 6A*).

Interestingly, a similar conceptual model can be applied to account for the rotation of the covariation plane across different gaits in humans: walking, hopping, running, air-stepping, obstacle clearance, crouched walking (*Ivanenko et al., 2007*). *Figure 6B* illustrates a superposition of $u_3$ vectors across animals (*Figure 4A*) and across different human gaits (indicated by colored confidence cones). Note a similar plane of rotation of the $u_3$ vector across animals and across different human gaits.

Therefore, limb kinematics of animal locomotion in the sagittal plane can be modeled by two principal components that determine limb segment coordinated movements (*Figures 3*, *4* and *6B*). The orientation of the first ($u_1$) and second ($u_2$) principal component axes on the covariation plane is illustrated for selected animals in the bottom panels of *Figure 6A* by blue and red lines, respectively. It has previously been argued that these components may be equivalent to the length and orientation of limb axis, and define an appropriate endpoint motion for different cat postures (*Lacquaniti and Maioli, 1994*) as well as human gaits (*Ivanenko et al., 2007*). One way to illustrate the functional significance of these principal components is to plot the changes in limb kinematics resulting by a corresponding shift along $u_1$ (PC$_1$) and $u_2$ (PC$_2$) axes (*Lacquaniti and Maioli, 1994*). Examples of stick diagrams generated by such shifts are shown in *Figure 6C*, demonstrating that PC$_1$ mainly reproduces the changes of limb orientation while PC$_2$ is mainly limited to changes in limb length.

## Concluding remarks

We showed that the planar covariation law previously established for humans holds for the terrestrial walk of several mammals and birds. This kinematic synergy lies at the interface between two highly conserved phenomena in animal locomotion, the neural command signals output by central pattern generators on the one hand, and the mechanics of the body COM. The kinematic synergy may therefore represent one specific neuromechanical principle of control of the instantaneous position of the COM, thus contributing to net mechanical energy savings. Our study was exploratory (although extremely laborious, see Materials and methods), since our results were mainly obtained under natural conditions rather than the controlled conditions of a laboratory. Nevertheless, the results we found are grounds for further, more systematic investigations into why the principle of planar covariation might be conserved across animal species.

The present findings suggest a modular control organization whereby appropriate coordination of the limb segments for each animal can be reduced to two independent components. The orientation of the covariation plane remarkably differed between HL and FL and between animals, especially for HL (*Figure 4A*). The major changes in HL plane across animals were associated with the second PC related to the limb length covariation ($u_2$ vector, *Figure 6A*), probably reflecting different distribution of stiffness and phase control of oscillations and thus the relative rotation of limb segments. Specific limb segment phase relationships are likely advantageous for each individual animal species, and might be the result of evolution. For instance, birds master flight techniques, perform take-off and landing manoeuvres, and accordingly they could adopt a specific yielding kinematic synergy (*Figure 3A*, *Figure 3—figure supplement 2*, *Figure 4A*). In quadrupeds, different biomechanical functions of HL and FL imply limb-specific couplings of neural oscillators with limb mechanical oscillators.

In sum, our study provides an integrative view on the dynamic template of limb segment motion across a wide range of animals and prompts further work to understand functional and evolutionary advantages of specific planar covariation patterns adopted by different species.

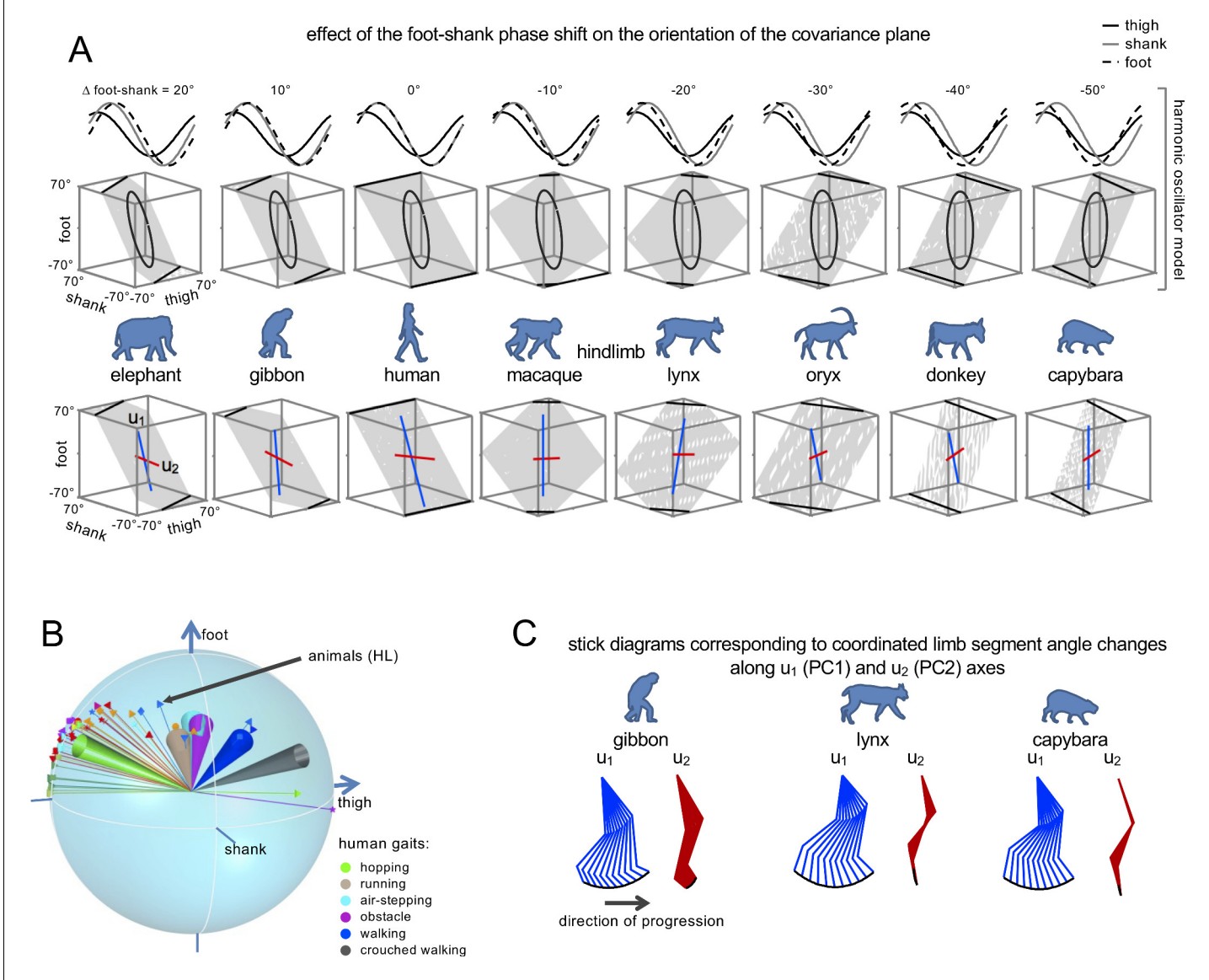

**Figure 6.** Inter-segmental coordination patterns. (**A**) Effect of the foot-shank phase shift on the orientation of the covariation plane. Upper panels: Harmonic oscillator model of *Barliya et al. (2009)* approximating the three segment elevation angles by sinusoidal waveforms with specific phase and amplitude. By changing the phase of the foot segment waveform relative to the shank segment from 20° to −50° (corresponding to the same range of $\Delta_{foot-shank}$ in *Figure 5A*), the covariation plane rotates similarly to the actual plane rotation across animals (lower panels, blue and red lines indicate the orientation of $u_1$ and $u_2$ vectors, respectively). (**B**) Similar plane of rotation of the $u_3$ vector across animals (HL, *Figure 4A*) and different human gaits (indicated by colored confidence cones, the data are redrawn from *Figure 2B* in *Ivanenko et al., 2007*). (**C**) Examples of stick diagrams corresponding to the coordinated limb segment angle changes along $u_1$ (PC1) and $u_2$ (PC2) axes (indicated by color lines in panel A, bottom plots). Note that PC1 reproduces the whole limb orientation changes while PC2 is mainly associated with changes in the limb length. Source files are available in the SourceData6-Figure6.zip file.

DOI: https://doi.org/10.7554/eLife.38190.023

The following source data is available for figure 6:

**Source data 1.** Inter-segmental coordination patterns.

DOI: https://doi.org/10.7554/eLife.38190.024

## Materials and methods

### Animals and protocols

For most species, a few different animals were recorded (*Table 1*). The recordings were made in different locations, most of them at Falconara (Italy) Zoo, others at Rome or Nemi (Italy) Zoo. The videos of the Sika deer were made at Nara city in Japan (where these animals are allowed to freely walk with humans). Human walking was recorded in the laboratory. For mice, videos came from a previously published study (Movie S1 in *Akay et al., 2014*), while for six therian mammals the kinematics was recorded by *Fischer et al. (2002)* using cineradiography to study the contribution of limb segment angles to step length and FL and HL movements. In the latter case, published graphs of limb segment elevation angles were scanned, digitized manually and time-interpolated to fit a normalized 100 points time base for the analysis of the inter-segmental coordination.

It is important to note that animals were observed walking spontaneously and at their preferred speed. Furthermore, most recordings were performed in natural outside conditions and on the terrain where each animal lives, with no laboratory stress or human handlers' interaction. No special permission is required in Italy for non-invasive observation of animals outside laboratory settings in behavioral studies like the present one (Italian law: DL 26/2014). As for the previously published studies, the mouse (*Akay et al., 2014*) and six therian mammals (*Fischer et al., 2002*) walked on a treadmill, but the operator adjusted the treadmill speed to obtain preferred speeds of the animals. *Table 1* reports the animals analyzed, their scientific name, the typical body weight reported from the literature, the speed, the Froude number (normalized speed, see below), and the number of recorded strides.

### Data collection

The recordings of animal walking were made using a Fujifilm Camera (FinePix SL1000, at 60 Hz, 37 species) or a Canon camera (EOS 550D, at 50 Hz, 10 species). Cameras were fixed on a tripod to limit vibrations during recordings, and were oriented roughly orthogonal to the direction of animal walk. The distance between the camera and the animals ranged between 4 and 10 m for all recordings, depending on where the animal walked with respect to the observation point of the experimenter. For humans, we recorded both overground (with the Fujifilm camera) walking in one subject and treadmill walking in five subjects (at 5 km/h, using a 9-camera Vicon-612 system, Oxford, UK, sampling rate 100 Hz).

From video recordings, we identified successful sequences of strides when the gait occurred in the sagittal plane steadily and on a straight path roughly perpendicular to the optical axis of the camera to minimize errors in 2-D kinematic analysis (*Kim et al., 2008*). Only complete strides were analyzed using hindlimb touchdown as the onset. We obtained the kinematics of both the right and left side by recording locomotion in both directions (relative to the camera), and the kinematic data were pooled together because both sides in walking have similar and repeatable locomotion characteristics (*Hildebrand, 1967*; *Alexander and Jayes, 1983*). The contralateral HL and FL endpoints were used only to characterize the interlimb coordination (diagonality of gait, see below). The number of recorded strides varied across animals; on average, we recorded 9 ± 10 (mean ± SD) successful strides per animal (452 strides total, *Table 1*).

### Kinematic data reconstruction

Once we selected the successful strides in videos, the reconstruction was performed using the Tracker software (v.4.95), a free video analysis and modeling tool built on the Open Source Physics Java framework. The anatomical landmarks of the ipsilateral side (with respect to the camera) tracked in the reconstruction were: hip (HIP), knee (KNE), and ankle (ANK) joints, base of the external metatarsal (or tarso-metatarsus in birds) (MT), endpoint (end of the distal phalanx) of the hindlimb (HEP). For quadrupedal animals, we also tracked the following forelimb landmarks: shoulder (SHO), elbow (ELB), and wrist (WRI) joints, base of the external metacarpal (MC), and endpoint (end of the distal phalanx) of the forelimb (FEP) (see *Figure 2—figure supplement 1A,B* left panels). In addition, we tracked the dorsal border of the scapular spine (SCA), nose (NOS), and tail endpoint (TEP) landmarks. To characterize the interlimb coordination (diagonality of gait), the contralateral hindlimb (C_HEP), and forelimb (C_FEP) endpoints were also tracked.

All anatomical landmarks were manually tracked frame by frame using a wireless touchpad with digital stylus (Wacom Bamboo Pad CTH-300), and using the skeleton model of each animal (as derived from the literature) for guidance (*Figure 2—figure supplement 1A,B* left panels). In total, 33,510 frames were processed by a very experienced person (author GC) in 7227 hr of work.

These kinematic data were further processed in the context of a multi-segmented bipedal (*Figure 2—figure supplement 1A*, left panel) and quadrupedal (*Figure 2—figure supplement 1B*, left panel) model. The analyses were performed using custom-made algorithms implemented in Matlab. The kinematic data were low-pass filtered using a zero-lag, fourth-order dual-pass Butterworth filter with a cutoff of 10 Hz. Next, we applied a custom model-based algorithm that uses the average segment length over all frames in each trial, and optimizes the locations of joint centres by constraining the changes in the limb segments lengths (*Catavitello et al., 2015*).

## Sensitivity analysis of the markerless approach

Since we recorded animal walking under natural conditions and we included animals that are usually difficult to train and work with (e.g. lion, cougar, tiger) or have very large sizes (e.g. ostrich, addax, giraffe, elephant, hippopotamus), we could only apply the markerless approach to reconstruct the kinematic data (*Catavitello et al., 2015*). We previously verified that this approach yields reliable results in the assessment of phase relationships between limb segment angles (*Catavitello et al., 2015*). In this previous study, we compared the results of the kinematic analysis of canine locomotion obtained from a video camera with those obtained with a high-performance 3D motion-capture system (SIMI Motion system, Unterschleissheim, Germany, sampling rate 100 Hz), and we found very similar characteristics of the inter-segmental coordination (*Catavitello et al., 2015*).

In the present study, we compared the results in five species (goose, pigeon, guinea fowl, elephant, and cat) with published data on elevation angles in the same or related species walking at comparable speeds. In the latter case, the data were obtained by means of high-performance 3D motion-capture systems (*Shen and Poppele, 1995*; *Ren et al., 2008*; *Stoessel and Fischer, 2012*). We scanned the graphs of limb segment elevation angles published in these reports, digitized them manually and time-interpolated to fit a normalized 100 points time base for the analysis of the inter-segmental coordination. We found a good agreement between our results and those obtained in related species in the previous publications. Thus, on average the root mean square (RMS) difference between the angular waveforms of our study and the corresponding ones of the previous studies was $5.5 \pm 3.1°$, and the average correlation coefficient between angular waveforms was 0.94 (obtained by pooling all segments and all steps together. Furthermore, the orientation of the covariation plane derived from both sets of studies was almost identical: the angular difference between $u_3$ vectors was on average 0.7° for HL and 0.5° for $FL_{low}$.

In addition, in the present study, we compared the results obtained from the video camera recordings of the human subject in the laboratory with those reported in the literature. At matched walking speed (mean=1.6 m/s across all strides), the characteristics of the planar covariation of the limb segment elevation angles were identical to those reported by Bianchi et al. (*Bianchi et al., 1998b*) (orientation of the covariation plane [see below]: $u_{3t}$=0.04, $u_{3s}$= −0.72, $u_{3f}$ =0.68).

Finally, while all anatomical landmarks were manually tracked by a single person (see above), we verified the inter-rater reliability for a few animals. To this end, we asked another researcher, very experienced in tracking kinematic data but unaware of the details of the present study, to track the videos of the avocet, camel, cheetah, gibbon and Sika deer. The mean RMS difference between the angular waveforms obtained by the two persons was $3 \pm 3°$ and the mean correlation coefficient was 0.99 (all elevation angles and all strides being pooled together). The corresponding difference in the orientation of the covariance plane ($u_3$) obtained by the two persons was low: on average 0.8° for HL, 2.0° for $FL_{upp}$ and 2.5° for $FL_{low}$.

## General gait parameters

The general model of the HL and FL segments is shown in *Figure 2—figure supplement 1A,B* (left panels). Whole limb and trunk orientations were defined from the HIP-HEP (HL), SCA-FEP (FL) and SCA-HIP (trunk) segments. The hindlimb was modeled as the multi-segmented limb from HIP to HEP, consisting of the thigh, shank, foot and toes segments, while the forelimb (from SCA to FEP) included the scapula, upper arm, lower arm, hand, and finger segments. Whole limb and limb

segment elevation angles relative to the vertical were calculated and analyzed, angles being positive when the distal marker was located anterior to the proximal marker. Mean trunk inclination was defined as the mean angle between the trunk and the horizontal reference (*Figure 2—figure supplement 2E*).

The gait cycle for each limb was defined as the time-interval between two successive maxima of the limb orientation waveform (*Catavitello et al., 2015*). The stance phase (when the foot was on the ground) corresponded to the time window between the maximum and the following minimum of the limb orientation waveform. We considered the gait cycle defined by the ipsilateral HL and FL (facing the camera). The contralateral HL and FL endpoints were used only to characterize the inter-limb coordination (diagonality of gait). The quadrupedal gait (inter-limb coupling) can be characterized by the footfall sequence (*Hildebrand, 1976*). To this end, the phase lag was computed as the relative timing ($t_{FL}$, $t_{FLcontr}$) of the FL cycle onset with respect to HL, and expressed as a percentage of the gait cycle (*Figure 2—figure supplement 2B*). The lateral gait is determined when the HL touchdown is followed by the ipsilateral FL touchdown, whereas in the diagonal sequence it is followed by the contralateral FL touchdown.

Limb endpoint excursion was determined separately for fore-aft and up-down (relative to the body) movements (*Figure 2—figure supplement 2F*). To compare different animals, the calculated values were normalized to hindlimb length (*L*), defined as the sum of the average lengths of thigh, shank, and foot segments over all frames in each video (except for humans, in which case we used the thigh + shank length, as it is more commonly accepted in the literature due to the heel contact with the ground).

To estimate walking speed *V*, we computed the distance covered by the hip landmark of a given animal during a stride. Since for most animals we could not measure the individual body length in meters, we approximated it using data from the literature in order to convert *V* from pixels/s to m/s. We also assessed the dimensionless walking speed (Froude number, *Fr*), which is suitable for the comparison of the speed of locomotion in animals of very different size (*Alexander, 1989*). The Froude number is given by $Fr = \frac{V^2}{g \cdot L}$, where *g* is the acceleration of gravity. The estimated Froude numbers are reported in Table 1.

## Tri-segmented limb model

We used a tri-segmented limb model (*Fischer and Blickhan, 2006*) and serially homologous HL and FL segments, starting from the distal segment: foot-hand, shank-lower arm, and thigh-upper arm. However, the scapula segment also undergoes significant rotations in the sagittal plane in most mammalian groups during locomotion. Accordingly, the tri-segmented model for HL included thigh (HIP-KNE), shank (KNE-ANK) and foot (ANK-MT) interconnected segments, while for FL we used two tri-segmental models (*Fischer and Blickhan, 2006*): 1) $FL_{upp}$ – scapula (SCA-SHO), upper arm (SHO-ELB), and lower arm (ELB-WRI), and 2) $FL_{low}$ – upper arm, lower arm, and hand (WRI-MC).

## Inter-segmental coordination

Our study was mainly focused on a general locomotor pattern in various species. Accordingly, the waveforms of the elevation angles of the limb segments were time interpolated over individual gait cycles to fit a normalized 100-point time base, and averaged first across strides and then across animals, in order to obtain a general template of HL and FL angular motion for each animal species. Nevertheless, even though the kinematic data tend to be repeatable across consecutive strides (*Faber et al., 2002*; *Kim et al., 2008*), we also report the inter-stride variability for a few animal species, in which we recorded more than 15 strides (dog, donkey, human, *Table 1*).

The inter-segmental coordination of the elevation angles of HL and FL segments was evaluated in position space using principal component analysis (PCA) as previously described (*Borghese et al., 1996*; *Bianchi et al., 1998b*; *Ivanenko et al., 2007*; *Catavitello et al., 2015*). To assess planar covariation of limb segment motion, we computed the covariance matrix of the ensemble of time-varying elevation angles (after subtraction of their mean values). The three eigenvectors $u_1$, $u_2$ and $u_3$, rank-ordered on the basis of the corresponding eigenvalues, correspond to the orthogonal directions of maximum variance in the sample scatter. The first two eigenvectors $u_1$ and $u_2$ identify the best-fitting plane of angular covariation. The third eigenvector ($u_3$) is the normal to the plane, and defines the plane orientation in the 3D space of the elevation angles. The planarity of the trajectories

was quantified by the percentage of total variation ($PV_3$) accounted for by the third eigenvector (for ideal planarity, $PV_3 = 0\%$).

Our main analysis was focused on the elevation angles that capture directly the limb configuration in external space. However, we also report the results of the PCA applied to the relative (anatomical) angles of the hip, knee and ankle for HL, and shoulder, elbow and wrist for FL.

## Orientation of the covariation plane

To characterize the distribution and differences in the orientation of the covariation plane ($u_3$) across groups (birds HL, mammals HL, mammals $FL_{upp}$, mammals $FL_{low}$), we used the empirical shape criterion (*Fisher et al., 1993*) and the high concentration parameter test (*Tsagris et al., 2017*). In particular, we distinguished the girdle distribution from the clustered distribution of the eigenvectors based on the empirical shape criterion. Briefly, let $(x_1, y_1, z_1), \ldots, (x_n, y_n, z_n)$ be the direction cosines of a sample of points on the unit sphere. The location of these points can be synthetized by their sample mean vector $(\bar{x}, \bar{y}, \bar{z})$, which is defined as $(\bar{x}, \bar{y}, \bar{z}) = (\sum x_i, \sum y_i, \sum z_i)$, where the sum is over the number of points $n$. It is useful to express the mean vector in polar form as $(\bar{x}, \bar{y}, \bar{z}) = R * (\bar{x}_0, \bar{y}_0, \bar{z}_0)$, the scalar product of a unit vector $(\bar{x}_0, \bar{y}_0, \bar{z}_0)$ with its resultant length $R$. If the points $(x_1, y_1, z_1), \ldots, (x_n, y_n, z_n)$ are considered as having equal mass, then their center of mass is $(\bar{x}, \bar{y}, \bar{z})$, which has direction $(\bar{x}_0, \bar{y}_0, \bar{z}_0)$ and distance $R$ from the origin. Then, the mean direction $(\bar{x}_0, \bar{y}_0, \bar{z}_0)$ defines the location of the sample, and the mean resultant length $\bar{R} = R/n$ provides a measure of how concentrated the sample is. If the points are concentrated close together, $R$ will be close to 1, whereas an increasing scatter results in smaller values of $R$. We then computed the eigenvalues $(\tau_1, \tau_2, \tau_3)$ of the orientation $T$ defined as $T = \begin{pmatrix} \sum x_i^2 & \sum x_i y_i & \sum x_i z_i \\ \sum x_i y_i & \sum y_i^2 & \sum y_i z_i \\ \sum x_i z_i & \sum y_i z_i & \sum z_i^2 \end{pmatrix}$. These eigenvalues give an indication of the general shape of the data set. The empirical shape criterion is based on the assumption that the shape of the distribution will determine its position in the bi-dimensional space created by the variables $a = \log(\tau_3 / \tau_2)$ and $b = \log(\tau_2 / \tau_1)$. For the girdle distribution, the empirical shape value (defined as $\gamma = a/b$) is less than 1, while for the clustered distribution it exceeds 1 (see Fig3.15b in *Fisher et al., 1993*). The degree of alignment of the samples of $u_3$ vectors was assessed by the concentration parameter $k$, which is a measure of the concentration of the sample about the mean direction (the higher is $k$, the more clustered the data), which we used to distinguish the population mean directions (*Tsagris et al., 2017*).

## Phylogenetic considerations of kinematic traits

We also added an analysis of kinematic traits taking into account a consideration of the data variation in an evolutionary context. In this context, species are not independent data points. Indeed, closely related species derive from a common ancestor and should be weighted more than distant species, since they share common characteristics (*Garland et al., 2005*). We performed a phylogenetic analysis of the orientation of the covariance plane for the HL ($\alpha$ angle, *Figure 4A*) and its correlation with the phase shifts between limb segment movements (*Figure 5A*). If traits under consideration of closely related species have similar values while the similarity decreases with increasing phylogenetic distance, one can infer the presence of a high phylogenetic signal of the trait. Vice versa, a weak phylogenetic signal corresponds to the situation when close relatives are not more similar on average than distant relatives. The latter may happen in the presence of a convergent evolution or when the traits are randomly distributed across a phylogeny.

We reconstructed a taxonomic tree with the data stored in the NCBI Taxonomy Browser (https://www.ncbi.nlm.nih.gov/Taxonomy/Browser/wwwtax.cgi). We used the Brownian motion model (random walk in continuous time) to estimate trait evolution: modifications of the trait value through time occur gradually and they are independent of the current state. At the tips of the phylogenetic tree, the expected covariance between trait values of the species is proportional to their common history. The history of each tip, that is species, is computed as the sum of their branch length. Then, a phylogeny can be represented as an $n \times n$ phylogenetic variance–covariance matrix, where $n$ is the number of species in the phylogeny. The diagonals of the matrix correspond to the total length of the tree and they represent the species variances, while the off-diagonal elements are computed as

the sum of their shared branch lengths and are the covariances between species pairs. We computed the branch lengths with the function 'compute.brlen' in the 'phytools' R package (*Revell, 2009*).

We computed Blomberg's K that measures phylogenetic signal by quantifying the amount of observed trait variance relative to the trait variance expected under Brownian motion (*Blomberg et al., 2003*). K is the ratio of the mean squared error of the tip data in relation to the phylogenetic mean of the data divided by the mean squared error extracted from a generalized least-squares model (PGLS, Phylogenetic Generalized Least-Squares) that uses the phylogenetic variance–covariance matrix in its error structure (*Kamilar and Cooper, 2013*). K can vary continuously from zero, indicating that there is no phylogenetic signal in the trait (i.e. the trait has evolved independently of phylogeny), to infinity. K = 1 indicates that there is strong phylogenetic signal and the trait has evolved according to the Brownian motion model of evolution, while K > 1 indicates that close relatives are more similar than expected under a Brownian motion model of trait.

We performed the evolutionary phylogenetic correlation by fitting a linear module using PGLS. The correlations were computed using the 'phytools' and 'geiger' R packages. The ancestral states estimation was computed assuming Brownian motion, using the functions 'fastAnc' and 'plotSimmap' in the 'phytools' R package (*Revell, 2009*).

## Statistics

Statistical analyses were performed using Matlab and R software. Descriptive statistics included the calculation of the mean values and standard deviations. For each species, the parameters were first averaged across gait cycles and then across animals for the same species before subsequent analyses. The general gait parameters (trunk orientation, stride and stance duration, inter-limb coupling, Froude number and endpoint excursions) were not computed for animals marked by an asterisk in *Table 1* (since this information was not provided in *Fischer et al. (2002)*), so we performed statistics on these parameters for a smaller number of animals (n = 48) as compared with the analysis of the inter-segmental coordination (n = 54). To assess some general kinematic parameters, paired t-tests were used to evaluate differences between HL and FL in quadrupeds, while unpaired t-tests were used to assess differences between the groups of animals when appropriate (e.g. birds vs. mammals HL or FL). Statistical analysis of spherical data was used to characterize the mean orientation of the normal to the covariation plane. In spherical statistics, we distinguished the girdle distribution from the clustered distribution of the eigenvectors with the empirical shape criterion (*Fisher et al., 1993*). Multi-way ANOVA and high concentration parameters test for spherical data were carried out with the R package built for directional statistics (*Tsagris et al., 2017*). A linear regression analysis was used to assess the relationship between limb parameters and rotation of the covariation plane. The correlation coefficients were Z-transformed before the statistical analysis. Reported results are considered statistically significant for p<0.05.

## Data availability

We provided Source Data files for figures. They are labeled as Source Data x-figure x.zip (where x is the figure number), contain numeric data and, where necessary, the code to reproduce the figure. See the readme file for each folder.

## Acknowledgements

We thank the Parco Zoo Falconara staff, particularly Iole Palanca, Gioia Gaiot, and Renato Piccinini, for help with video recordings and Dr. Giovanni Martino for help with tracking kinematic data. This research was financially supported by the Italian Ministry of Health (IRCCS Ricerca corrente), Italian Space Agency (grant n. I/006/06/0), Italian Ministry of University and Research (PRIN grant 2015HFWRYY_002), and Horizon 2020 Robotics Program (ICT-23–2014 under Grant Agreement 644727-CogIMon).

## Additional information

### Funding

| Funder | Grant reference number | Author |
|---|---|---|
| Ministero della Salute | IRCCS Ricerca corrente | Francesco Lacquaniti |
| Agenzia Spaziale Italiana | I/006/06/0 | Francesco Lacquaniti |
| Ministero dell'Istruzione, del-l'Università e della Ricerca | PRIN grant 2015HFWRYY_002 | Francesco Lacquaniti |
| Horizon 2020 | Robotics Program ICT-23-2014 under Grant Agreement 644727-CogIMon | Yury Ivanenko Francesco Lacquaniti |

The funders had no role in study design, data collection and interpretation, or the decision to submit the work for publication.

### Author contributions

Giovanna Catavitello, Conceptualization, Resources, Data curation, Software, Formal analysis, Validation, Investigation, Visualization, Methodology, Writing—original draft, Project administration, Writing—review and editing; Yury Ivanenko, Conceptualization, Formal analysis, Supervision, Visualization, Methodology, Writing—original draft, Project administration, Writing—review and editing; Francesco Lacquaniti, Conceptualization, Supervision, Funding acquisition, Visualization, Writing—original draft, Project administration, Writing—review and editing

### Author ORCIDs

Giovanna Catavitello (iD) http://orcid.org/0000-0002-2286-7321
Yury Ivanenko (iD) http://orcid.org/0000-0001-9363-9548
Francesco Lacquaniti (iD) http://orcid.org/0000-0003-0896-7315

### Decision letter and Author response

Decision letter https://doi.org/10.7554/eLife.38190.027
Author response https://doi.org/10.7554/eLife.38190.028

## Additional files

### Supplementary files

• Transparent reporting form
DOI: https://doi.org/10.7554/eLife.38190.025

### Data availability

All data analysed during this study are included in the manuscript and supporting files. Source data files have been provided.

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
