## [Decision Letter]

Thank you for submitting your article "A kinematic synergy for terrestrial locomotion shared by mammals and birds" for consideration by *eLife*. Your article has been reviewed by three peer reviewers, including Ronald L Calabrese as the Reviewing Editor and Reviewer #3, and the evaluation has been overseen by a Reviewing Editor and Eve Marder as the Senior Editor. The following individuals involved in review of your submission have agreed to reveal their identity: Raeed Chowdhury (Reviewer #2).

The reviewers have discussed the reviews with one another and the Reviewing Editor has drafted this decision to help you prepare a revised submission.

Summary:

In this manuscript, the authors present a kinematic analysis of 54 different species (10 birds and 44 mammals including man) performing free walking locomotion. The different animals present large differences in body size, mass (ranging from 0.030 Kg to 4,000 Kg), limb configuration, and amplitude of angular movements. Despite these differences, PCA analysis of covariation of the temporal changes of the elevation angles of the limbs during both stance and swing shows that ~95% or more of the variance is explained by two principal components so that covariation is coplanar in the three dimensional space of joint angles. In the Discussion section, the authors argue quite convincingly from simple modeling that these two components correspond to whole limb orientation changes and changes in limb length respectively. They also argue that the orientation of the covariation plane differed according to different phase relationships between elevation angles, and the third principal component of the covariation matrix being orthogonal to the best fitting plane characterizes its orientation. This orientation is correlated with the phase shift Δ_foot-shank_. The phase tuning of the planar covariation is relatively constant across mammals (tetrapods) at the forelimbs, whereas the phase at the hind limbs varies greatly across species, consistent with biomechanical and functional differences of the distal segments. Because this kinematic synergy lies at the interface between the motor pattern produced by locomotor pattern generators, the mechanics of the body center of mass and the external environment, it may represent one neuromechanical principle conserved in evolution to contribute saving mechanical energy.

The paper is clearly written and is nicely illustrated. It contains a wealth of comparative data, and its comprehensive comparative analyses and elegant conclusions, combined with this rich data set, assures that it will be of wide interest.

Essential revisions:

1) Chief among concerns is that the authors are performing so many statistical tests (correlations) on features of the data that it seems obvious that some of the tests will result in significant correlations. A typical defense against these potential false positives is to limit the number of tests and present only those driven by hypotheses based on physiological principles (or pilot data). However, that's not necessary in an exploratory study like this, where the goal is to find signs for conserved principles of gait across species. Instead, the authors should acknowledge in the discussion the exploratory nature of this study, focusing on the fact that the correlations found are grounds for further investigation into why these principles might be conserved, rather than a true set of rules for terrestrial gait.

2) Introduction: We are not convinced that the covariation plane implies that the limb length and orientation are being specifically controlled. On the subject of limb endpoint being encoded in the dorsal spinocerebellar tract, one recently published paper (Chowdhury, Tresch and Miller, 2017) suggests that the endpoint tuning described by Bosco and Poppele, (2001) can actually be a simple consequence of the musculoskeletal geometry of the hindlimb. Other studies in motor control of reaching suggest that neurons that look like they encode endpoint kinematics can be explained by the biomechanics of the limb (Lillicrap and Scott, 2013). This paragraph doesn't seem to be crucial to the manuscript and should be deleted or reframed to take into account the cited literature.

3) Although the literature typically uses the angle of each segment to a vertical line, but we wonder what these results would look like if the analyses were performed by using the actual joint angles, i.e. the angle between a segment and its neighbor. The results might be similar to those already described, but it would find it very useful for the manuscript to at least briefly describe what those results look like.

4 Add a consideration of size and phylogeny to the analysis, which might reveal novel patterns to help us understand the proposed conserved nature of neuromechanical control? See Revell, 2009. The careful consideration of the variation in an evolutionary context might reveal hints at control similar to the way that development and pathologies have tested the control law. Moreover, it is generally accepted that phylogenetic controls help avoid phylogenetic pseudoreplication (the exaggeration of the statistical significance of a set of measurements resulting from treating species data as independent observations when they are in fact interdependent because of their evolutionary history). Phylogenetic corrections can significantly change the view of relationships among diverse species (Garland, Bennett and Rezende, 2005). Put simply, many closely related species should be weighted more as a single data point, so as to not over-represent their statistical significance.

In the present data, the assumption of independent evolution of all species might not change the results, given the relatively small variation. Ogihara et al., (2104) have previously concluded that, "The planar covariation of inter-segmental coordination has evolved independently in both avian and human locomotion, despite the different mechanical constraints".

5 The Introduction is very clear except on one point. The third paragraph attempts to present planar covariation succinctly, but I believe this is not enough for a general reader or even one like myself who follows the motor pattern generation literature. Please indicate what is meant more precisely by planar covariation, perhaps even referring to PCA and the 3D space of joint angles.

Reviewer #2:

- Subsection “Planar covariation of limb segment elevation angles”: The orientation of the covariance plane depends entirely on how you orient your axes. In the figure, the scapula is the up/down axis for the upper forelimb, but the hand is the up/down axis for the lower forelimb, and it's unclear why those particular orientations are chosen.

- Subsection “Planar covariation of limb segment elevation angles”: I would have liked to see an explanation of why the forelimb was separated into two separate models, when PCA could have worked equally well in four dimensions. My assumed explanation is that this makes it easier to compare the synergies between forelimb and hindlimb, but I don't think this (or any other reasoning) was in the manuscript.

---

## [Author Response]

Summary:In this manuscript, the authors present a kinematic analysis of 54 different species (10 birds and 44 mammals including man) performing free walking locomotion. The different animals present large differences in body size, mass (ranging from 0.030 Kg to 4,000 Kg), limb configuration, and amplitude of angular movements. […] The paper is clearly written and is nicely illustrated. It contains a wealth of comparative data, and its comprehensive comparative analyses and elegant conclusions, combined with this rich data set, assures that it will be of wide interest.

We thank the reviewers for their evaluation of the findings and helpful constructive suggestions served to improve the manuscript. We tried to incorporate them all.

Essential revisions:1) Chief among concerns is that the authors are performing so many statistical tests (correlations) on features of the data that it seems obvious that some of the tests will result in significant correlations. A typical defense against these potential false positives is to limit the number of tests and present only those driven by hypotheses based on physiological principles (or pilot data). However, that's not necessary in an exploratory study like this, where the goal is to find signs for conserved principles of gait across species. Instead, the authors should acknowledge in the discussion the exploratory nature of this study, focusing on the fact that the correlations found are grounds for further investigation into why these principles might be conserved, rather than a true set of rules for terrestrial gait.

We decreased the number of tests. In particular:

- We eliminated the statistical tests on the FL_low_ and FL_upp_ ROMs, as suggested by the reviewers in their specific comments.

- We also omitted statistics on the planarity index.

As suggested, we acknowledged the exploratory nature of our study. We now explicitly stated it in the Concluding remarks, as well as the suggestion for further investigations.

“Our study was exploratory (although extremely laborious, see Materials and methods), since our results were mainly obtained under natural conditions rather than the controlled conditions of a laboratory. Nevertheless, the results we found are grounds for further, more systematic investigations into why the principle of planar covariation might be conserved across animal species”.

We also mentioned:

“In sum, our study provides an integrative view on the dynamic template of limb segment motion across a wide range of animals and prompts further work to understand functional and evolutionary advantages of specific planar covariation patterns adopted by different species.”

2) Introduction: We are not convinced that the covariation plane implies that the limb length and orientation are being specifically controlled. On the subject of limb endpoint being encoded in the dorsal spinocerebellar tract, one recently published paper (Chowdhury, Tresch and Miller, 2017) suggests that the endpoint tuning described by Bosco and Poppele, (2001) can actually be a simple consequence of the musculoskeletal geometry of the hindlimb. Other studies in motor control of reaching suggest that neurons that look like they encode endpoint kinematics can be explained by the biomechanics of the limb (Lillicrap and Scott, 2013). This paragraph doesn't seem to be crucial to the manuscript and should be deleted or reframed to take into account the cited literature.

We deleted this paragraph, as suggested. This issue needs further investigations. Even though the biomechanics of the limb and musculoskeletal geometry contribute to the endpoint tuning, the CNS should take these geometrical considerations into account for the control of foot trajectory and limb movements.

3) Although the literature typically uses the angle of each segment to a vertical line, but we wonder what these results would look like if the analyses were performed by using the actual joint angles, i.e. the angle between a segment and its neighbor. The results might be similar to those already described, but it would find it very useful for the manuscript to at least briefly describe what those results look like.

We quantified the planar covariation also for the relative joint angles.

We added to the Results section:

“We examined mainly the intersegmental coordination of the elevation angles, rather than that of the relative joint angles (so called anatomical angles), because the former capture the overall limb configuration in external space. Indeed, the elevation angles identify the orientation of each segment relative to the direction of gravity (vertical direction). Moreover, the time course of the anatomical angles of flexion-extension in human locomotion is more variable across subjects and trials than that of the elevation angles, and the planarity of the anatomical angles trajectories is weaker (Borghese et al. 1996; Barliya et al. 2009).In our recorded animals, when we applied the PCA to the anatomical angles (hip, knee and ankle for HL, and shoulder, elbow and wrist for FL), the planarity indexes (PV3) were: 2.7 ± 2.3% for HL angles (ranging from 0.02% in guinea fowl to 8.0% in porcupine) and 4.0 ± 2.5% for FL angles (ranging from 0.7% in ox to 13.9% in porcupine). Therefore, although also the anatomical angles trajectories tend to be constrained close to one plane, the planar coordination of the elevation angles is stronger and less variable, in agreement with what was previously reported for human walking (Borghese et al. 1996; Barliya et al. 2009).”

and to the Materials and methods section:

“Our main analysis was focused on the elevation angles that capture directly the limb configuration in external space. However, we also report the results of the PCA applied to the relative (anatomical) angles of the hip, knee and ankle for HL, and shoulder, elbow and wrist for FL.”

4 Add a consideration of size and phylogeny to the analysis, which might reveal novel patterns to help us understand the proposed conserved nature of neuromechanical control.? See Revell, 2009. The careful consideration of the variation in an evolutionary context might reveal hints at control similar to the way that development and pathologies have tested the control law. Moreover, it is generally accepted that phylogenetic controls help avoid phylogenetic pseudoreplication (the exaggeration of the statistical significance of a set of measurements resulting from treating species data as independent observations when they are in fact interdependent because of their evolutionary history). Phylogenetic corrections can significantly change the view of relationships among diverse species (Garland, Bennett and Rezende, 2005). Put simply, many closely related species should be weighted more as a single data point, so as to not over-represent their statistical significance.In the present data, the assumption of independent evolution of all species might not change the results, given the relatively small variation. Ogihara et al., (2104) have previously concluded that, "The planar covariation of inter-segmental coordination has evolved independently in both avian and human locomotion, despite the different mechanical constraints".

As suggested, we added a consideration of size and phylogeny to the analysis of the orientation of the covariance plane (α parameter, Figure 4) and its correlation with phase shifts between limb segment movements (Figure 5).

We added a new figure (Figure 4—figure supplement 1) and the corresponding text in the Results section:

“We searched for the presence of a phylogenetic signal in the wide scatter of α-values of rotation for HL across animal species, in order to frame the data scatter in an evolutionary context. Although the K index (Blomberg et al., 2003, see Materials and methods section) we used for the presence of a phylogenetic signal in the α-angle for HL was statistically significant, its value was rather low (K=0.10, n=54, p=0.04) (Figure 4—figure supplement 1), suggesting that the pattern of α-angles distribution is hardly dependent on phylogenetic relatedness of the species considered. Such a pattern may occur when close relatives are less similar than distant ones.”

Also subsection “Relationship between limb parameters and rotation of the covariation plane”:

“After controlling for a potential phylogenetic signal in the response (and, hence, non-independence of the residuals, see Materials and methods), we found that the α-angle remained significantly correlated with Δ_foot-shank_ (r^2^=0.37, p<0.00001) and Δ_upper arm-scapula_(r^2^=0.31, p<0.00001).”

and in the Discussion section:

“We also searched for the presence of a phylogenetic signal in the scatter of α-values of rotation for HL across animal species, and we found only a weak one. Indeed, the α-values of close relatives were not more similar between each other than to the values of distant ones (Figure 4—figure supplement 1). This suggests that the planar covariation is a feature that has arisen independently several times during evolutionary history (Ogihara et al., 2014). A convergent evolution of this kinematic synergy may be due to both adaptation and constraints acting similarly in distantly related species. Adaptation would arise due to the advantage of a kinematic control law lying at the interface between neural commands and environment. Constraints would depend on the inherent biomechanical coupling between different limb segments.”

and in subsection “Phylogenetic considerations of kinematic traits”:

“We also added an analysis of kinematic traits taking into account a consideration of the data variation in an evolutionary context. In this context, species are not independent data points. Indeed, closely related species derive from a common ancestor and should be weighted more than distant species, since they share common characteristics (Garland et al. 2005). We performed a phylogenetic analysis of the orientation of the covariance plane for the HL (α angle, Figure 4A) and its correlation with the phase shifts between limb segment movements (Figure 5A). If traits under consideration of closely related species have similar values while the similarity decreases with increasing phylogenetic distance, one can infer the presence of a high phylogenetic signal of the trait. Vice versa, a weak phylogenetic signal corresponds to the situation when close relatives are not more similar on average than distant relatives. The latter may happen in the presence of a convergent evolution or when the traits are randomly distributed across a phylogeny.

[…]

We performed the evolutionary phylogenetic correlation by fitting a linear module using PGLS. The correlations were computed using the “phytools” and “geiger” R packages. The ancestral states estimation was computed assuming Brownian motion, using the functions “fastAnc” and “plotSimmap” in the “phytools” R package (Revell, 2009).”

5 The Introduction is very clear except on one point. The third paragraph attempts to present planar covariation succinctly, but I believe this is not enough for a general reader or even one like myself who follows the motor pattern generation literature. Please indicate what is meant more precisely by planar covariation, perhaps even referring to PCA and the 3D space of joint angles.

As suggested, we briefly introduced the PCA approach in the Introduction: “Kinematic coordination of limb segments can be described by statistical methods such as principal component analysis (PCA), which projects movements onto a low-dimensional space thereby helping to detect invariant properties of coordination (Daffertshofer et al., 2004). Based on this approach, one law of inter-segmental coordination has been described in human locomotion, which involves the planar covariation of the temporal changes of the elevation angles of the lower limbs (Borghese et al., 1996). Specifically, limb segment rotations covary so that the three-dimensional (3D) trajectory of the elevation angles lies close to a plane.”

Reviewer #2:- Subsection “Planar covariation of limb segment elevation angles”: The orientation of the covariance plane depends entirely on how you orient your axes. In the figure, the scapula is the up/down axis for the upper forelimb, but the hand is the up/down axis for the lower forelimb, and it's unclear why those particular orientations are chosen.

We thank the reviewer for pointing to this typo. We corrected the labels of Figure 3A.

- Subsection “Planar covariation of limb segment elevation angles”: I would have liked to see an explanation of why the forelimb was separated into two separate models, when PCA could have worked equally well in four dimensions. My assumed explanation is that this makes it easier to compare the synergies between forelimb and hindlimb, but I don't think this (or any other reasoning) was in the manuscript.

We thank the reviewer for this suggestion, indeed using a tri-segmented model makes it easier to compare the synergies between forelimb and hindlimb. As suggested, we added to the Results section:

“We used serially homologous HL and FL segments and models for comparing the kinematics of the HL and FL, starting from the distal segment: foot-hand, shank-lower arm, and thigh-upper arm. However, the scapula segment also undergoes significant rotations in the sagittal plane (Figure 2). While PCA can be applied also in 4 dimensions for FL, using a tri-segmental model makes it easier to compare the kinematic synergies between FL and HL. Therefore, for FL we used two separate tri-segmental models (Fischer and Blickhan 2006): FL_low_ – 'upper arm–lower arm–hand’ and FL_upp_ – ‘scapula–upper arm–lower arm’ (Figure 3A right panel).”

We also made a separate subsection: “Tri-segmented limb model”.